# Hydrogen evolution with hot electrons on a plasmonic-molecular catalyst hybrid system

Ananta Dey[1], Amal Mendalz[1], Anna Wach [2,3], Robert Bericat Vadell [1], Vitor R. Silveira [1], Paul Maurice Leidinger [2], Thomas Huthwelker[2], Vitalii Shtender [4], Zbynek Novotny [2], Luca Artiglia [2] & Jacinto Sá [1,5] ✉

Plasmonic systems convert light into electrical charges and heat, mediating catalytic transformations. However, there is ongoing controversy regarding the involvement of hot carriers in the catalytic process. In this study, we demonstrate the direct utilisation of plasmon hot electrons in the hydrogen evolution reaction with visible light. We intentionally assemble a plasmonic nanohybrid system comprising $NiO/Au/[Co(1,10\text{-Phenanthrolin-5-}amine)_2(H_2O)_2]$, which is unstable at water thermolysis temperatures. This assembly limits the plasmon thermal contribution while ensuring that hot carriers are the primary contributors to the catalytic process. By combining photoelectrocatalysis with advanced in situ spectroscopies, we can substantiate a reaction mechanism in which plasmon-induced hot electrons play a crucial role. These plasmonic hot electrons are directed into phenanthroline ligands, facilitating the rapid, concerted proton-electron transfer steps essential for hydrogen generation. The catalytic response to light modulation aligns with the distinctive profile of a hot carrier-mediated process, featuring a positive, though non-essential, heat contribution.

Plasmonic photocatalysis uses electrical charges formed during the plasmon resonance decay triggered by light absorption. A plasmon is a quantised oscillation of the electron density, and its decay can generate high-energy carriers (electrons and holes) in metals due to the interaction between plasmons and incident light, causing them to become 'hot' or have high kinetic energy[1]. These hot carriers can then be used to generate electrical current or to drive chemical reactions. However, hot electrons' involvement in catalysis remains disputed, despite reports of their participation in processes such as solar to chemical energy reactions[2–5], epoxidations[6,7], dehydrogenations[8], ammonia electrosynthesis[9], etc[10–12]. The scepticism surrounding their involvement relates to the hot carriers' ultrafast relaxation (ca. 1 ps)[13], and that several examples rationalise their participation as enhancers of the photothermal process. Therefore, the catalytic output is prone to errors since the surface temperature of plasmonic materials is notoriously tricky to measure accurately, thus underestimating the thermal contribution to the catalysis[14,15]. Despite the significant progress, it remains challenging to disentangle charge carrier catalysis from photothermal effects[16–19].

The hot carriers' energy distribution is broad, with a significant fraction of the carriers having energies above the Fermi level of the metal caused by the non-Fermi-Dirac distribution[20]. More research is needed to fully understand plasmonics' hot carrier energy distribution dynamic behaviour[21,22]. Still, the ultrafast relaxation can be partially mitigated via ultrafast charge transfer to suitable acceptors consecutively[23,24], or simultaneously[25], forming this contribution scientific basis to demonstrate the direct involvement of hot carriers in the catalytic process.

Herein, a plasmonic nanohybrid system consisting of $NiO/Au/[Co^{II}(phen\text{-}NH_2)_2(H_2O)_2]$ (phen-$NH_2$ = 1,10-Phenanthrolin-5-amine) was

[1]Department of Chemistry-Ångström, Physical Chemistry division, Uppsala University, Box 532, 751 20 Uppsala, Sweden. [2]Paul Scherrer Institut, CH-5232 Villigen PSI, Switzerland. [3]SOLARIS National Synchrotron Radiation Centre, Jagiellonian University, Krakow, Poland. [4]Department of Materials Science and Engineering, division of Applied Materials Science, Uppsala University, 75103 Uppsala, Sweden. [5]Institute of Physical Chemistry, Polish Academy of Sciences, Marcina Kasprzaka 44/52, 01-224 Warsaw, Poland. ✉e-mail: jacinto.sa@kemi.uu.se

assembled and demonstrated to perform hydrogen evolution reaction (HER). NiO acted as a hole acceptor[26–29], and the cobalt complex, a mimic of the HER catalyst reported by Luo et al.[30] and the hydrolytic DNA cleavage agent by Sharma et al.[31], as an electron acceptor. Water can be converted into hydrogen through thermolysis that requires temperatures of 500–1000 °C[32], a temperature at which our molecular system would decompose, suggesting the participation of hot electrons in the catalytic process. To further this finding, the reaction mechanism was monitored by combining photoelectrocatalysis, unbiased ultrafast spectroscopies, and in situ electrochemistry, followed by near ambient pressure X-ray photoelectron spectroscopy (NAP-XPS) studies. The results suggest a reaction mediated by the phenanthroline-ligands that accept the electrons from the plasmon and transfer them to the cobalt centre in two concerted proton-electron transfers (CPET) that significantly lowers the energy threshold of the steps as it avoids the formation of higher energy intermediates[33]. Recently, a study was published with a similar concept, namely a cobalt porphyrin supported on plasmonic nanoparticle that, on illumination, produced $H_2$[34]. Still, there is a clear distinction. In the present contribution, only the Au nanoparticles (Au NPs) are photoactive, contrasting with the published study where the catalyst and Au NPs are photoactive. Thus, their observation might be related to photonic enhancement instead of a plasmonic hot carrier. Moreover, the authors suggested a cooperative result between plasmon hot carriers and localised thermal effects, for which this study does not have evidence. This contribution also offers more extensive experimental support for the mechanism reaction involving the plasmon hot carrier and Co complex catalyst ligands, which are markedly different from what has been published on cobalt systems for HER. The proposed combined spectroscopically approach offers a robust methodology to measure the reaction mechanism on mesoporous electrodes that represent real electrodes more truthfully.

## Results and discussion

### Electrodes fabrication and characterisation

The molecular catalyst mimics the system proposed by Sharma et al.[31] as a DNA hydrolytic cleavage agent. The prime difference between Sharma's system and the one presented herein is the presence of the amine group on the ligand, which is necessary to anchor the catalyst to the Au NPs. Consequently, the as-prepared catalyst has a cobalt centre coordinated with two 1,10-Phenanthrolin-5-amine ligands and a bidentate nitrate group. A second out-sphere nitrate ensures complex neutrality (Fig. 1B), which is consistent with previously reported crystal structures[31]. Details on the catalyst preparation and characterisation can be found in supporting information (SI). The optical spectrum of the as-prepared catalyst in dimethylformamide is shown in Fig. S4. It displays a strong absorption peak centred at 290 nm with a shoulder at 360 nm, characteristic of phenanthroline complexes[35]. The molecular catalyst thermally decomposes at 265 °C, significantly below the temperature needed for water thermolysis.

Cobalt nitrate complexes are known to exchange their nitrate ligands with water[36] which is the solvent used to attach the complex to the Au NPs and perform the catalysis. Therefore, the complex was titrated with water to evaluate whether ligand exchange occurred. Figure 1A shows the increase of the UV-Vis shoulder located at 360 nm, with an increase in water content, saturating at around 20% water content. The exchange was also confirmed by the X-ray photoelectron spectroscopy (XPS) analysis. The N $1s$ region in Fig. S13, acquired in the

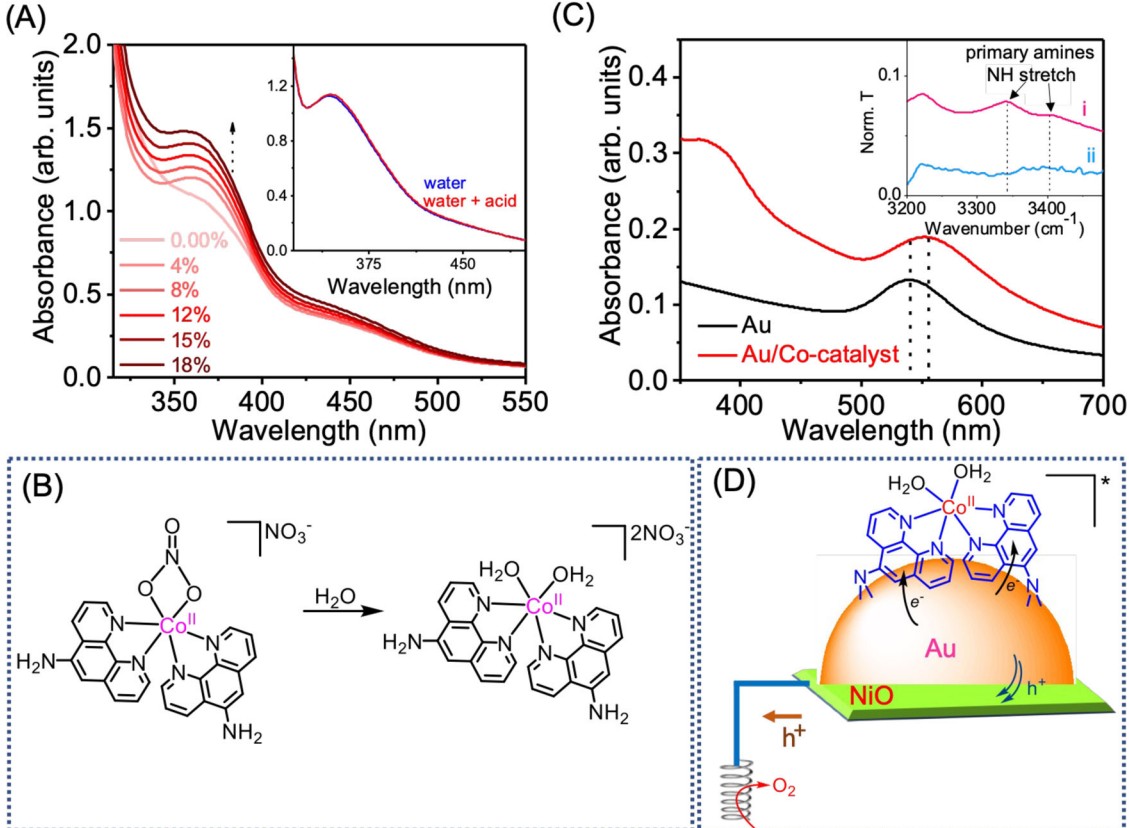

**Fig. 1 | Standard characterisation of the cobalt catalysts and photosystem.**
**A** Cobalt complex in dimethylformamide titration with water followed by in situ UV-Vis, with inset showing the effect of acid in the spectrum, it represents with and without acid in water; **B** proposed structure of the catalyst in water, which was used to anchor the catalyst to Au NPs; **C** UV-Vis spectra of Au NPs region before and after addition of the cobalt complex on thin film. On inset infrared spectra of the amino stretching region normalised by C-N stretching band intensity: (i) catalyst before anchoring; (ii) catalyst after anchoring it to the Au NPs; **D** photosystem structure used for the photo-electrocatalytic $H_2$ evolution.

low vacuum after introducing the electrode in the Near Ambient Pressure (NAP) analysis chamber, displays a sharp peak centred at 398.6 eV. Such a binding energy value can be assigned to the pyridinic nitrogen of the phenanthroline ligand[37]. Nitrate ligands are typically found at a binding energy of 408 eV[38], but there were no features in this region in the collected spectrum.

Further elaboration on ultra-high vacuum (UHV)-XPS analysis will be provided when corroborating evidence regarding anchoring the catalyst to the Au surface is presented. Notably, adding acid to the aqua complex did not change its optical absorption (Fig. 1A inset), suggesting that the di-aqua complex resultant from the exchange of the nitrate ligand by water molecules is very stable. Figure 1B shows the proposed catalyst structure after ligand exchange.

The attachment of the complex was followed by UV-Vis and infrared spectroscopies (Fig. 1C). The Au NPs were prepared using the Turkevich method, as reported elsewhere[39] and briefly described in SI. The UV-Vis of the Au NPs on glass shows the characteristic localised surface plasmon resonant (LSPR) peak at 535 nm, corresponding to an average particle size of $8 \pm 2$ nm (determined by dynamic light scattering (Fig. S5) and atomic force microscopy (AFM) (Fig. S6)), consistent with what we have published previously[40]. Note that NiO morphology (Fig. S7) and electronic structure (Fig. S8) did not change with Au NPs deposition and subsequent annealing.

The Au LSPR peak shifts to lower energy when the cobalt catalyst is added (Fig. 1C), confirming the anchoring. Note that the LSPR peak absorption is sensitive to the surrounding dielectric medium[41]. Consequently, the surface modification by the catalyst should induce a shift in the LSPR maximum as absorbed. Additionally, it is possible to see the complex absorption shoulder located at 370 nm, corroborating the attachment between the catalyst and Au NPs. Unfortunately, the glass support (FTO or cover glass) covers the rest of the complex UV-Vis band, precluding their measurement.

The 10-Phenanthrolin-5-amine ligand was purposely chosen to ensure selective coordination to the gold surface via the amino groups.

This first supporting evidence came from XPS measured at low vacuum conditions, with a Co $2p$ signal related to the catalyst only when Au NPs were present. The Co $2p_{3/2}$ on Au/Co-cat and NiO/Au/Co-cat measured had a single contribution centred at around 780.5 eV, consistent with Co is oxidation +2[42]. The observation that the Co signal was only present when Au is present is a solid endorsement for the selective anchoring of the catalyst to the gold surface. The anchoring is believed to occur via the -NH$_2$ groups[43]. This was corroborated by the disappearance of the amino bands in the infrared (Fig. 1C inset). Before anchoring, the complex has three small peaks between 3450–3300 cm$^{-1}$ and 3250–3200 cm$^{-1}$ associated with N-H stretching modes of primary amino groups[44]. The correspondent bending modes between 1650–1580 cm$^{-1}$ are also visible but somewhat overlapped by the water O-H bending mode. After attaching the catalyst to the Au NPs, the N-H bands disappear. The complete disappearance suggests that the catalyst coordinates to the Au NPs via both 1,10-Phenanthrolin-5-amine ligands, as shown schematically in Fig. 1D. The infrared bands were normalised to the C-N stretch at 1280 cm$^{-1}$ intensity to enable direct comparison. The C-N band is unaffected by the attachment, making it suitable for normalisation. Unfortunately, the formed Au-N bonds are not infrared active, and the low loading prevented their detection with Raman spectroscopy. Therefore, UHV-XPS experiments were performed to substantiate the catalyst anchoring to the Au surface via the -NH$_2$ group.

The UHV-XPS comparing the N $1s$ and Au $4f$ signals before and after anchoring the catalyst to the Au surface are presented in Fig. 2A, B, respectively. Before attaching, the catalyst has two N $1s$ peaks: the N from the phenanthroline bonded to the cobalt centre at 398.8 eV[45,46] and N at 401.4 eV ascribed to the -NH$_2$ groups[47]. Note that the UHV-XPS also did not show a peak ascribed to the nitrate groups, corroborating its exchange by water molecules. Upon attaching, the N $1s$ signal related to the -NH$_2$ group disappeared, and the N from phenanthroline shifted to 399.1 eV and got broader (FWHM before 1.641 and after attaching 1.812). Note that one expects the N $1s$ from the amino group

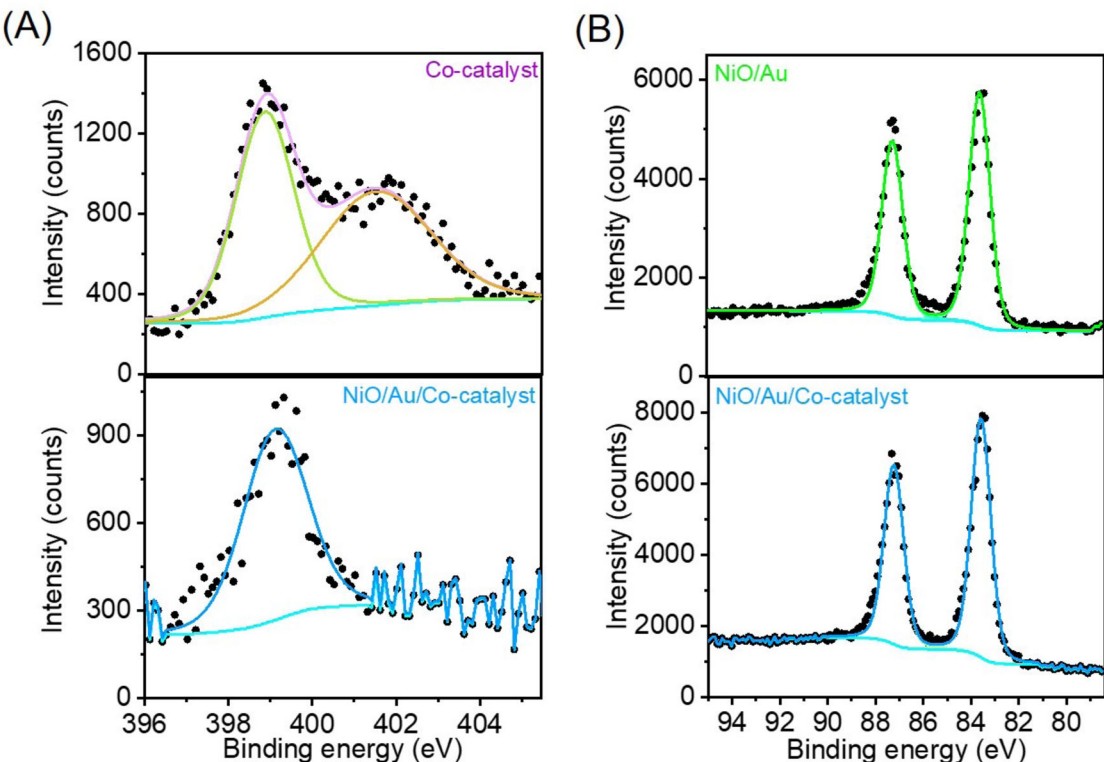

**Fig. 2 | Catalyst anchoring to Au NPs.** UHV-XPS shows the changes due to the catalyst after anchoring it to the Au surface via the -NH$_2$ functional groups. **A** XPS N $1s$ and **B** Au $4f$ regions.

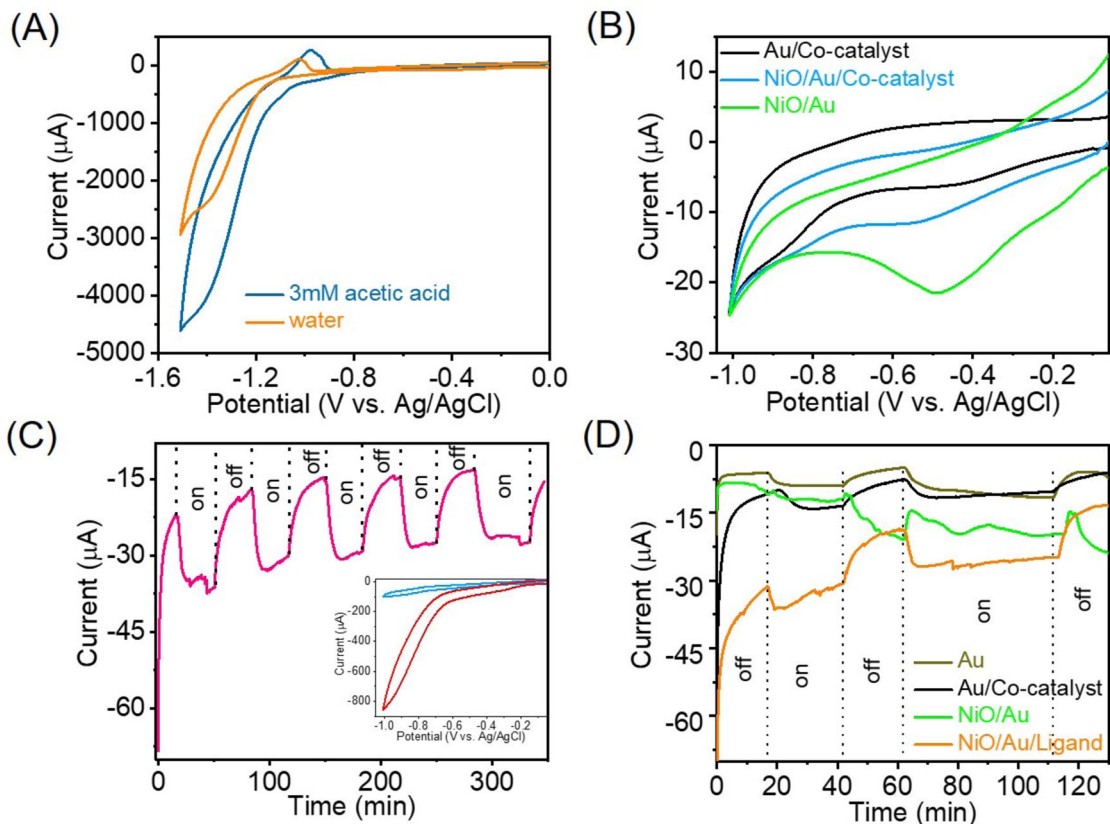

**Fig. 3 | Electrochemical and photo-electrocatalytic studies. A** Bulk electrolysis of cobalt catalysts in water and the presence of 3 mM acetic acid, using glassy carbon as the working electrode, Pt wire as the counter electrode, Ag/AgCl as reference electrode, 0.1 M LiCl as supporting electrolyte (pH = 5.2) and scan rate 50 mV/s; **B** Cyclic voltammetry of the NiO/Au and NiO/Au/Co-catalyst in water using Pt wire as a counter electrode, Ag/AgCl as reference electrode and 0.1 M LiCl as supporting electrolyte (pH = 5.2) (scan rate 50 mV/s); **C** Effect of light in the chron-oamperometry of the complete photosystem applying −0.65 V potential with the

inset showing the catalytic wave when the experiments were performed with 3 mM acetic acid (pH = 3.5). A 532 nm CW laser with 43.8 mW/cm² power was used as illumination. In inset NiO/Au/Co-catalyst in water (blue trace) and NiO/Au/Co-cat-alyst + acetic acid (red trace); **D** Effect of light in the chronoamperometry of the NiO/Au and NiO/Au/ligand applying −0.65 V potential in 3 mM acetic acid and 0.1 M LiCl (pH = 3.5). A 532 nm CW laser with 43.8 mW/cm² power was used as illumination.

to shift to lower binding energies as it loses the protons at a rate of about 1 eV per hydrogen atom lost[48]. These observations are consistent with attachment via the NH₂ groups and formation of Au-N species, which cannot be distinguished from the N of the phenanthroline ligand because of signal-to-noise. The Au *4f₇/₂* had binding energy at 83.6 eV for the sample with and without a catalyst, consistent with metallic gold[49]. There was only one species in all the samples, which is coherent with the idea that any electronic change due to NiO and Co-catalyst is delocalised over all the gold atoms, suggesting good electronic coupling[50]. Figure 1D shows the schematic representation of the complete photosystem.

### Photoelectrochemical studies and catalytic behaviour

Bulk electrolysis of the cobalt catalyst in water using 0.1 M LiCl as a supporting electrolyte without acid (Fig. 3A) shows no unique reduc-tion peaks before the onset of the catalytic wave at −1.18 vs Ag/Ag/Cl. Initiating with the absence of reduction peaks, this discovery diverges from the observations made by Luo et al.[30] (whose system served as inspiration for this study) and the system studied by Wang et al.[51,52] exhibiting comparable rigidity and coordination. The onset potential for electrolytic hydrogen production without using acid in our catalyst was lower, differing by only ca. 15 mV from Wang et al.[52] system but by a noteworthy 190 mV from the Luo et al.[30] system.

Predictably, the catalytic wave's amplitude is heightened with the presence of protons, underscoring proton availability as a pivotal factor in accessing catalytic performance within this system.

Consequently, all catalytic data was obtained in a 3 mM acetic acid environment. Although the introduction of acid did not result in new reduction peaks, it did cause a notable shift in the onset potential, reducing it by as much as 120 mV - a deviation from the characteristics of the previously documented system. These observations imply the engagement of two concerted proton-electron transfer mechanisms as opposed to the conventional sequential reduction followed by protonation[51]. Subsequent sections will delve into additional evidence supporting this proposed mechanism.

Figure 3B shows the cyclic voltammetry of the NiO/Au and NiO/Au/Co-catalyst thin films in water. The NiO/Au shows a reduction peak centred at around −0.49 V vs Ag/AgCl related to the reduction of gold surface oxygen and weak catalytic wave staring at −0.90 V vs Ag/Ag/Cl. Adding the cobalt catalyst drastically reduced the peak at −0.49 V vs Ag/Ag/Cl, suggesting that surface functionalisation by the catalyst decreases the amount of adsorbed oxygen on Au NPs. The cobalt loading in the complete system was determined to be 11.4 µg/cm² by inductively coupled plasma-optical emission spectrometry (ICP-OES), equating to 0.29 wt.% of Co. The low loading creates issues regarding spectroscopy signal-to-noise but ensures that the activity is primarily due to the catalyst and enables the detection of catalyst degradation evidence.

The complete system is susceptible to illumination at 532 nm, as shown by the CV in Fig. S17. Illumination of the entire system at plas-mon resonance led to a slight decrease in offset potential (ca. 0.05 V) but, more importantly, an increase in the catalytic wave photocurrent.

Chronoamperometry data at −0.65 V vs Ag/Ag/Cl in the absence and presence of light (CW laser 532 nm, selectively exciting only the Au plasmon) is presented in Fig. 3C. The photosystem is responsive to the light, increasing the photocurrent by ca. −15 µA (−19 µA/cm²). The $H_2$ production rate was estimated from three consecutive on/off light switches to be 3.1 nmol/(min.cm²). The increase in photocurrent was shown to be due to the evolution of $H_2$, as confirmed by online quadrupole mass spectrometry (QMS) analysis (Fig. S10). The response was found to be constant during the cycling of light on and off for the duration of the experiment (ca. 360 min). The findings indicate that the photosystem is relatively stable, and the process is catalytic. The significant increase in evolved $H_2$ relates to the presence of cobalt catalyst since neither the NiO/Au nor NiO/Au/ligand systems produce significant photocurrent (Fig. 3D) and had no detectable hydrogen evolution by online QMS analysis. The drift in the baseline is related to heating effects from the illumination and plasmonic decay because it lacked synchronisation with the light on/off cycles.

It is clear from the data that the systems with plasmonic materials are responsive to the 532 nm illumination. Light-mediated plasmon-catalysis is a very complex process due to many potential reaction enhancers. One possibility is the near-field enhancements caused by the local electric fields formed upon Au LSPR excitation. At the most basic level, near-fields can enhance charge separation and alignment of molecular dipoles[53]. However, such localised electric fields impact the hot carriers but cannot catalyse the reaction autonomously. The second option employs the plasmon-induced resonance energy transfer (PIRET) process, connecting the plasmon evanescent field to a semiconductor absorber through dipole-dipole interaction. However, these systems necessitate core-shell architectures (which are not applicable in this context)[54,55] and the most substantial enhancements were observed with silver as the plasmonic material rather than gold[56]. The final option explores strong-correlated plasmon-molecule systems. Still, in this scenario, there must be an optical overlap between the plasmon and molecule[50] which is again not present.

Catalytic performance measurements were conducted using off-resonant excitation to assess local field enhancement contribution. Off-resonance excitation induces local effects such as elevated local near-fields[57–59]. However, in the context of hot carriers, off-resonance excitation generates low-energy carriers that are not conducive to driving photocatalytic processes[60]. Excitation at 650 nm (off-resonant) caused no significant differences in the CV compared to experiments performed in the dark (Fig. S17). This result was further supported by light switch chronoamperometry (Fig. S18). The findings imply that local near-fields do not contribute significantly to enhancing catalysis. Consequently, the observed increase in catalytic output under resonant illumination is likely associated with hot electrons and heat rather than near-fields. This, however, does not rule out the potential for near-fields to assist catalysis by enhancing charge separation; they would likely influence hot carriers indirectly engaged in the catalytic process.

Heat is an inherent factor in plasmonic catalysis due to the underutilisation of hot charges, leading to their recombination and local heat generation. Although it is acknowledged that the surface temperature of excited plasmonic materials exceeds that of the solution, determining the precise value poses a challenge due to the ultrafast dynamics of thermalisation. The molecular catalyst remains stable only up to 265 °C, a temperature considerably lower than required for uncatalysed water thermolysis. Nevertheless, a noteworthy temperature range remains unexplored, primarily because experiments are conducted in an aqueous medium.

To disentangle the heat contribution, we performed light modulation chronoamperometry. A study by Maley et al.[61] demonstrated that light absorption at the electrode surfaces within nanoparticle arrays led to significant localised temperature increases and altered solution flows. These thermal effects were anticipated to influence electrochemical currents through diverse mechanisms, encompassing enhancements in mass transfer, shifts in equilibrium redox potentials, and conventional temperature-dependent accelerations in kinetic rates for electrode processes. Their analysis suggests that mass transfer enhancements alone would result in substantial current increases applicable to electrochemical reactions involving dissolved reactants and products, both outer-sphere and inner-sphere reactants. Consequently, heat-induced effects exhibit a distinctive gradual rise and decay of the current during light modulation, as they operate on processes with time constants in the nanosecond range.

The light response of the entire system (NiO/Au-Co-catalyst) in comparison to the system without NiO (i.e., Au/Co-catalyst) offers insights into the role of heat in the process (see Fig. 3C, D, compounded figure below). Removing NiO is expected to decrease the lifetime of the charge-separated state, generating more heat. However, contrary to the expectation that heat is the primary contributor to reactivity, we observed a fourfold increase in current induced by light when NiO was present. Additionally, examining the response of the Au/Co-catalyst to light modulation reveals a classic heat-mediated process with a relatively slow rise and decay, in contrast to the complete system. The entire system demonstrates a faster rise and decay to light modulation, indicating the involvement of hot carriers.

Light absorption by planar electrodes randomly decorated with plasmonic structures acts as a uniform heat source delocalised across the electrode–solution interface, resulting in heat dissipation in a linear geometry with significant temperature changes as a function of time. Consequently, the electrochemical response to light modulation provides a strategy to decouple heat from hot carriers' contributions to substantiate our mechanistic claim. Commonly, the experiments are performed by modulating the light intensity. However, the tested electrodes are quasi-transparent, making light-intensity modulation studies challenging. Thus, we opted to change the light ON/OFF cycle repetition rate to modulate electrode exposure to light.

Figure 4A shows the changes in measured photocurrent ($\Delta i$) as a function of light modulation repetition rate. Unsurprisingly, lower repetition rates (i.e. higher light exposure) resulted in more significant $\Delta i$. In a heat-mediated process, $\Delta i$ is expected to scale with $t^{1/2}$ ($t$ = time)[61] inconsistent with the observed current transients, providing the first substantiation for a hot carrier-mediated process. Additionally, $\Delta i$ in a heat-mediated electrochemical process follows a linear dependence with increased light exposure, independent of the process occurring via inner or outer-sphere reaction. Fig. 4B shows that $\Delta i$ does not offer a linear behaviour regarding light exposure, thus providing more explicit evidence for hot carriers' involvement.

**Elucidation of ultrafast charge dynamics**

Having established that the photosystem is responsive to light and able to evolve hydrogen, it is essential to determine whether plasmon hot electrons are involved in the process. Transient absorption spectroscopy (TAS) without an external electric field (unbiased) was performed to evaluate charge transfer in the photosystem. Excitation of the Au NPs LSPR results in a bleach signal and two small winglets on each side of the bleach to broaden the LSPR peak[62] (see the representative spectrum in SI Fig. S11). Figure 5A shows the kinetic traces extracted at 490 nm (edge of the positive winglet to the blue of the LSPR maximum) after excitation at 550 nm of the Au NPs, NiO/Au and NiO/Au/Co-catalyst systems. Note that similar findings were obtained when performing the analysis on the winglet to the red of LSPR maximum. The kinetic traces were fitted with a rising edge and a double exponential decay. The rising edge is assigned to the electron-electron (e-e) scattering lifetime, the shorter exponential decay to electron-phonon (e-ph) scattering lifetime and the longer decay to photo-phonon (ph-ph) scattering lifetime[13,14,63]. Recently, we demonstrated that charge transfer can be established from changes in the e-ph lifetime. Both electron and hole transfer reduce the e-ph lifetime

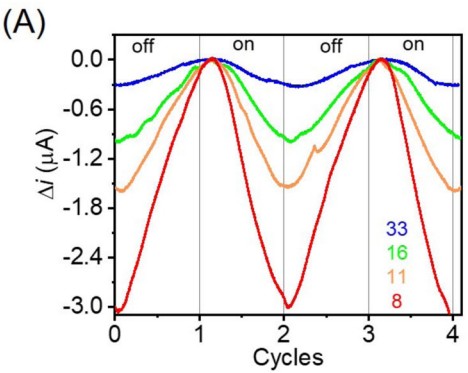
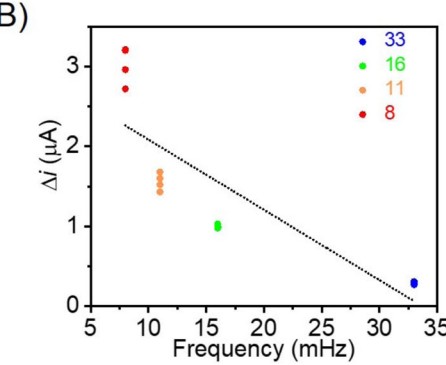

**Fig. 4 | Light-modulated photo-electrocatalytic studies.** The effect of light modulation in the chronoamperometry was performed at −0.65 V vs Ag/AgCl with 3 mM acetic acid (pH = 3.5) and 532 nm CW laser with 43.8 mW/cm². The experiments were performed using a squared function at different repetition rates. **A** Changes in photocurrent (Δ$i$) at different light modulation frequencies (8, 11, 16 and 33 MHz) over different light ON/OFF cycles; and **B** Δ$i$ versus light modulation frequency.

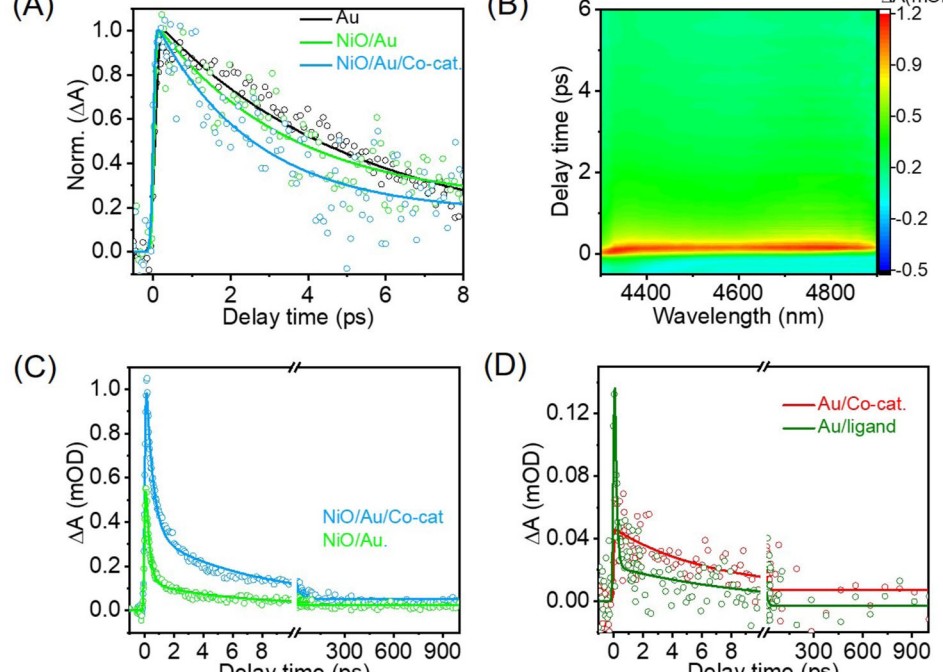

**Fig. 5 | Unbiased transient spectroscopy data after LSPR excitation at 550 nm.**
**A** Kinetic traces of Au, NiO/Au, NiO/Au/Co-catalyst, extracted at 490 nm from the TAS measurements; **B** TIRAS map of the NiO/Au/Co-catalyst; **C** TIRAS kinetic trace extracted at 4705 nm for NiO/Au and NiO/Au/Co-catalyst; **D** TIRAS kinetic trace extracted at 4705 nm for Au/ligand and Au/Co-catalyst.

compared to the plasmon nanoparticles without charge transfer. Hot electrons reduce the e-ph by taking energy from the resonance. Hot holes decrease the e-ph by injecting cold electrons into the resonance, reducing the average electron temperature[24,25].

The Au NPs on glass have an e-e lifetime estimated to be 167 ± 49 fs and an e-ph of 5.1 ± 0.4 ps, which is within what has been published[22,64,65]. When attached to NiO (hole acceptor), the Au NPs e-e increased slightly to 198 ± 85 fs with a noticeable decrease in e-ph lifetime to 3.4 ± 1.0 ps, consistent with what is anticipated if holes are transferred from Au NPs to NiO. The system composed of Au/Co-catalyst has an e-e of 148 ± 30 fs and a significantly shorter e-ph lifetime (4.2 ± 0.3 ps) compared with Au NPs alone. Since the catalyst is expected to be the electron acceptor, the reduction in e-ph lifetime suggests that electrons are transferred from Au NPs to the catalyst. The complete photosystem had an e-e of 198 ± 116 fs and the most

significant reduction in the e-ph from 5.1 ± 0.4 ps (Au NPs) to 2.6 ± 1.0 ps, suggesting the hot holes and electrons are transferred to the respective acceptors. In sum, the presence of electron and hole acceptors reduced the e-ph, consistent with charge transfer from Au NPs to the acceptors, with the most significant e-ph lifetime reduction observed when both electron and hole acceptors are present.

Simply formulated, the decrease in e-ph lifetime relates to changes in electron average temperature, which are regulated by the total energy in the resonance. Therefore, the observed reduction in e-ph lifetime can be related to an energy transfer[56]. To ascertain that the changes observed with TAS measurements are related to hot carrier transfer, not energy transfer, complementary unbiased transient infrared absorption spectroscopy (TIRAS) spectroscopy studies were performed. Free carriers absorb strongly in the infrared domain due to forming a quasi-metallic state[66]. The signal is characterised by broad

and featureless infrared absorption, often depicted as a background shift in the infrared spectrum[67].

A representative TIRAS data map after LSPR excitation at 550 nm is shown in Fig. 5B. Kinetic traces extracted between 4645–4700 nm (2150–2130 cm$^{-1}$) are presented in Fig. 5C. The kinetic traces were fitted with a rising edge and two exponential decays, ascribed to the lifetimes of the injection and recombination processes, respectively. NiO/Au shows a rising edge with a 196 ± 104 fs time component, suggesting fast injection of the hole. Most injected charge recombines within 417 ± 117 fs (-91%). The complete system displayed a similar injection time (100 ± 23 fs) and increased recombination time (6.6 ± 4.2 ps, 85% of the signal). However, an increase in signal amplitude is noticeable, suggesting that more charge is transferred when both acceptors are present, i.e., an enhancement of charge separation, resulting in more charge available for the catalysis. In both cases, 5–10% of the charge survives past 1 ns, making it useful for catalytic transformations, including H$_2$ evolution.

The free carrier signal detected by unbiased TIRAS confirms the transference of hot carriers to the acceptors, suggesting that the observed decrease in e-ph lifetime is less likely to be due to energy transfer. Since the Au plasmon resonance absorption is very far from the acceptors' absorptions, one can also discard the hypothesis of photonic enhancement as the corporate for catalytic performance. Noteworthy is that unbiased TIRAS measurements on a system without the NiO, namely with Au/Co-catalyst and Au/ligand, also show a broad featureless infrared absorption (Fig. S12), characteristic of free carrier absorption, not localised charge. The kinetic traces in Fig. 5D are noisy due to low signal and thus challenging to fit. However, qualitatively, the signal shows a short rising component (faster than the instrument response function (ca. 100 fs)), suggesting a high-speed electron injection. In the case of the Au/ligand, the decay is swift, but when Co is present, the decay is comparatively slow, with the charge surviving past 1 ns. The shape of the TIRAS signal indicates that it relates to free carries. This suggests that hot electrons are injected into the ligands due to the strong coupling between phenanthroline ligands and Au NPs; the charge is delocalised through the aromatic rings, acting as free carriers. The observation confirms ligand reduction under light irradiation occurs even without external bias, which is crucial for the hot electron-mediated mechanism.

The presence of Co improves hot electrons' lifetime due to some charge stabilisation. Still, the cobalt centre is the same since this would lead to the disappearance of the 'free carrier' infrared absorption behaviour, which only happens for at least 1 ns. The TIRAS observation is also consistent with the electrochemistry's absence of cobalt reduction peaks. This is a peculiar observation because, from the speculated mechanisms of the analogous[30] and relatable cobalt complexes[68,69] gives a central role to the metal centre, often undergoing sequential reductions and protonations before evolving hydrogen. Conversely, the ligands are confined to a spectator role in the catalytic process.

## In situ near ambient pressure (NAP)-XPS studies

To corroborate the peculiar finding, electrochemical experiments combined with in situ near ambient pressure (NAP)-XPS measurements were performed using the dip-and-pull approach[70] in the absence (Fig. S14) and presence (Fig. 6) of acetic acid. The measurements were performed with the mesoporous films used for photocatalytic data. This significantly reduces the signal intensity and requires adaptation of the dip-and-pull method to reduce the amount of electrolyte trapped in the porous film. Briefly, only the lowest segment of the sample was dipped into the electrolyte solution, while the rest formed a liquid film via capillary forces. After some equilibration time (ca. 30 min), the liquid film settled to a point where the photoemission signals of the electrode (O 1s, Co 2p and N 1s) were detectable together with the signal of liquid and gas phase water (O 1s). All spectra were corrected for charge and baseline, and the peaks were deconvoluted according to XPS fitting constraints[71]. Charging correction was performed using an advantageous C 1s signal.

Figures 6A and S8A show the O 1s spectra acquired in situ at different applied potentials. Three prominent peaks, centred at approximately 531.0 eV, 532.7 eV and 534.9 eV, are assigned to adsorbed hydroxyls, liquid water (thin electrolyte film on top of the electrode) and gas phase water, respectively[72,73] (fitting parameters are reported in Table S1). The slight shift of the O 1s central peak measured between the pure electrolyte and the same in the presence of acid (from 532.68 to 532.75 eV under OCP conditions) may be due to the presence of acetic acid, whose contribution falls within the spectral range of liquid water peak[74]. A fourth peak, centred at 529.5 eV, was

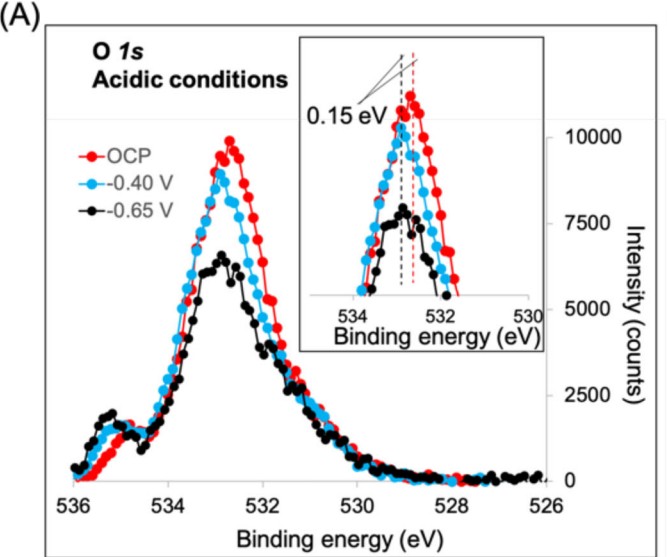

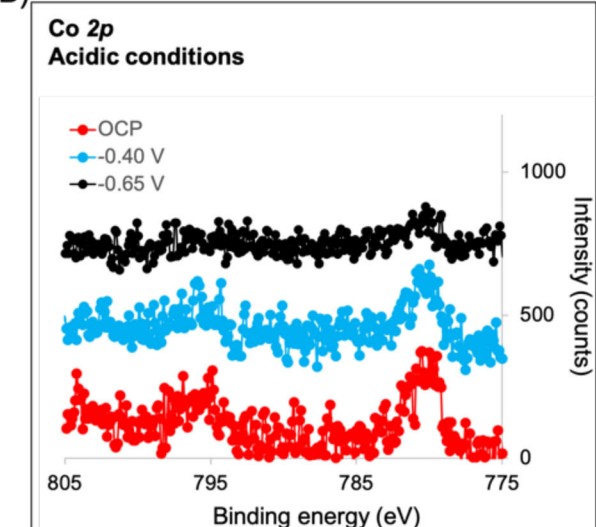

**Fig. 6 | In situ NAP-XPS of NiO/Au/Co-catalyst under variable potential in 3 mM acetic acid with an X-ray photon energy of 5000 eV. A** O 1s signals (the inset magnifies the prominent peaks, highlighting the binding energy shift due to the potential applied); **B** Co 2p signals overlaid with a 400 counts offset for better visibility.

detected in the absence of acid and assigned to lattice oxygen (electrode). Such a difference between the two conditions (no lattice oxygen detected in the presence of acid) suggests that the liquid electrolyte film in the absence of acid was thinner than in the presence of acid.

Potential control and the availability of a continuous liquid film up to the position monitored by XPS were surveyed by shifting the O $1s$ signal according to the applied potential. Figures 4A and S8A show a shift in the central peak of O $1s$, centred at approximately 532.8 eV and assigned to liquid water (thin electrolyte layer), as the potential was applied (detected as a positive binding energy shift, proportional to the potential applied to the WE), confirming experiment validity and thus access to Co oxidation state at different potentials. A difference of about 0.1–0.15 V was detected between the applied potential chosen based on the catalysis and seen in the O$1s$ NAP-XPS, which is assigned to calibration shifts of the reference electrode during long experimental times. Therefore, the values specified in the plots are applied for consistency, not measured voltages. It is essential to highlight that while the 532.7 eV peak component displays a shift proportional to the applied voltage, components at 529.5 and 531.0 eV do not.

Confident that NAP-XPS experiments reflected the Co oxidation state at different potentials, one can proceed with the analysis of the Co $2p$ region (Fig. 6B). Despite the low signal-to-noise ratio, due to Co $2p$ attenuation through the liquid electrolyte layer stabilised on the WE, the main features of Co $2p$ peaks can be detected on both investigated electrodes (see Figs. 6B and S14B). The Co $2p_{3/2}$ prominent peak is centred at around 780 eV, ascribed to $Co^{2+}$ as expected[75,76]. The peak position does not shift with the applied potential, as the working electrode is set to ground potential during the experiment. Nevertheless, a peak shift due to the formation of a different cobalt species is not observed either in the presence or in the absence of acetic acid (Fig. S8B). Indeed, the reduction of cobalt from 2+ to the metallic state leads to a shift of the binding energy of the $2p_{3/2}$ peak by about 2 eV, from 780 to 778 eV[77,78]. Figure 4B also shows that at −0.65 V vs Ag/AgCl, the cobalt signal decreases in intensity. Such a potential is sufficient for the evolution of $H_2$ since bubbles were detected at the electrode and marked the onset of the catalytic wave (Fig. S9). Thus, the decrease in signal-to-noise ratio relates to experimental conditions since the signal statistics are affected by the formation of H2 bubbles at the highest potential. Therefore, only $Co^{2+}$ is present until $H_2$ formation; thus, the reduction peak at lower voltages relates to the reduction of the phenanthroline ligands (Fig. S9). Since cobalt reduction is required to evolve hydrogen, not seeing its drop suggests that the cobalt centre reduction and protonation steps are fast and just before the hydrogen evolution, i.e., changes at the cobalt not rate limiting and cannot be detected due to the NAP-XPS temporal resolution[79].

Similar results occur in the absence of acid, but the onset potential of $H_2$ evolution was shifted to 0.8 V vs Ag/AgCl, i.e., about 0.15–0.2 V, as observed in the catalysis (Fig. S9). Furthermore, the Co $2p$ signal intensity detected was lower. The following hypotheses can be postulated to explain such a behaviour: (i) the liquid electrolyte film was thicker in this case, attenuating more the signal of the substrate; (ii) the surface of the electrode is slightly different in the absence of the acid, suggested by the more prominent electrode-related shoulder (centred at ca. 530.5 eV) in the O $1s$ spectra (Fig. S8).

## Mechanism proposal

Figure 7 shows a schematic representation of the two hypothetic catalytic cycles that can justify all the observations. Excitation of Au plasmon results in the formation of hot electrons and holes as part of its resonance decoherence via Landau damping. The hot holes are transferred to the NiO and react at the counter electrode, leading to $O_2$ production. The electrons are transferred to the phenanthroline ligands. After both ligands acquire a charge (i.e., undergo reduction), two potential mechanisms are hypothesised: a simultaneous

mechanism (blue arrow pathway) and a sequential mechanism (pink arrow pathway).

In the simultaneous mechanism, the catalyst experiences two rapid concerted proton-electron transfer (CPET) steps in the concurrent scenario. These steps lead to the reduction and protonation of the cobalt catalyst, ultimately resulting in $H_2$ evolution. The swift nature of this mechanism poses a challenge as it prevents the detection of any changes in the Co $2p$ signal in NAP-XPS. On the other hand, the sequential mechanism involves sequential CPET processes. The initial CPET yields a $Co^{III}$-H hydride intermediate with an electronic structure similar to $Co^{II}$. This interpretation aligns with the NAP-XPS observations and offers a more chemically sound reaction mechanism. Earlier investigations into Fe systems have identified analogous quasi-isoelectronic configurations[80,81] associated with the single-electron reduction of the metal centre and the addition of a proton. This process effectively results in the formation of a hydride, accompanied by an increase in the metal centre's oxidation state by +1. Nevertheless, further investigations are required for confirmation, though this is beyond the scope of the current contribution.

In summary, a photosystem was proposed to confirm the direct involvement of hot electrons in a photocatalytic process, in this case, the $H_2$ evolution process. The photosystem effectively mitigates the heat contribution by designing a catalyst that decomposes well below water thermolysis conditions, positioning heat as a mere enhancement rather than the primary cause for the observed $H_2$ evolution. Off-resonance measurements conclusively eliminate near-field contributions as the direct catalyst of the reaction, although they may still play a role in extending the lifetime of hot electrons. The catalytic response to light modulation exhibits a shape consistent with the desired electron mechanism, contrasting with the detrimental impact of heat. Nevertheless, it was discovered that heat does have some positive influence on catalysis. Unbiased ultrafast spectroscopic measurements confirm charge transfer to respective acceptors.

Additionally, in conjunction with NAP-XPS under variable potential, a postulated reaction mechanism highlights the crucial role of cobalt catalyst ligands. These ligands accept plasmon hot electrons and, through CPET steps, reduce and protonate the metal centre, ultimately leading to hydrogen evolution. This study conclusively resolves the longstanding debate within the research community regarding the direct involvement of hot carriers in the photocatalytic process.

## Methods
### Sample preparation
**Au nanoparticles (NPs) synthesis.** The Au NPs were synthesised following the Turkevich method, as published by Piella et al.[39]. Briefly, sodium citrate tribasic dihydrate (Merck, ACS reagent ≥ 99%) 50 mL (6.6 mM) water solution was taken in a 100 mL round bottom flask and stirred at 70 °C in an oil bath. Then, 0.1 mL (2.5 mM) tannic acid (Merck, ACS reagent ≥ 99.5%) was added to the reaction mixture. Finally, 1 mL of (25 mM) $HAuCl_4$ (Merck, ≥ 99.9%) was added instantly. After 5 min, the reaction mixture changed from dark blue to wine. The colour change confirms the formation of the Au nanoparticles. The synthesised Au nanoparticles were stored in a fridge. The size of the Au nanoparticles was analysed using dynamic light scattering (DLS).

**Cobalt catalyst synthesis.** 200 mg (1.02 mmol) 1,10-Phenanthrolin-5-amine (Merck, 97%) and 148.42 mg (0.51 mmol) cobalt(II) nitrate hexahydrate (Merck, ACS reagent ≥ 98%) were refluxed in 20 ml ethanol (Merck, ≥ 99.9% (GC)) for one hour. After that, the reaction mixture was filtered. The filtrate was placed in a clean beaker at room temperature for a few days without further disturbance, and we got a red precipitate. Finally, the obtained product was collected through filtration and dried in a vacuum desiccator. The yield 216 mg (73.9%). ATR-IR 3342 $cm^{-1}$, 3402 $cm^{-1}$.

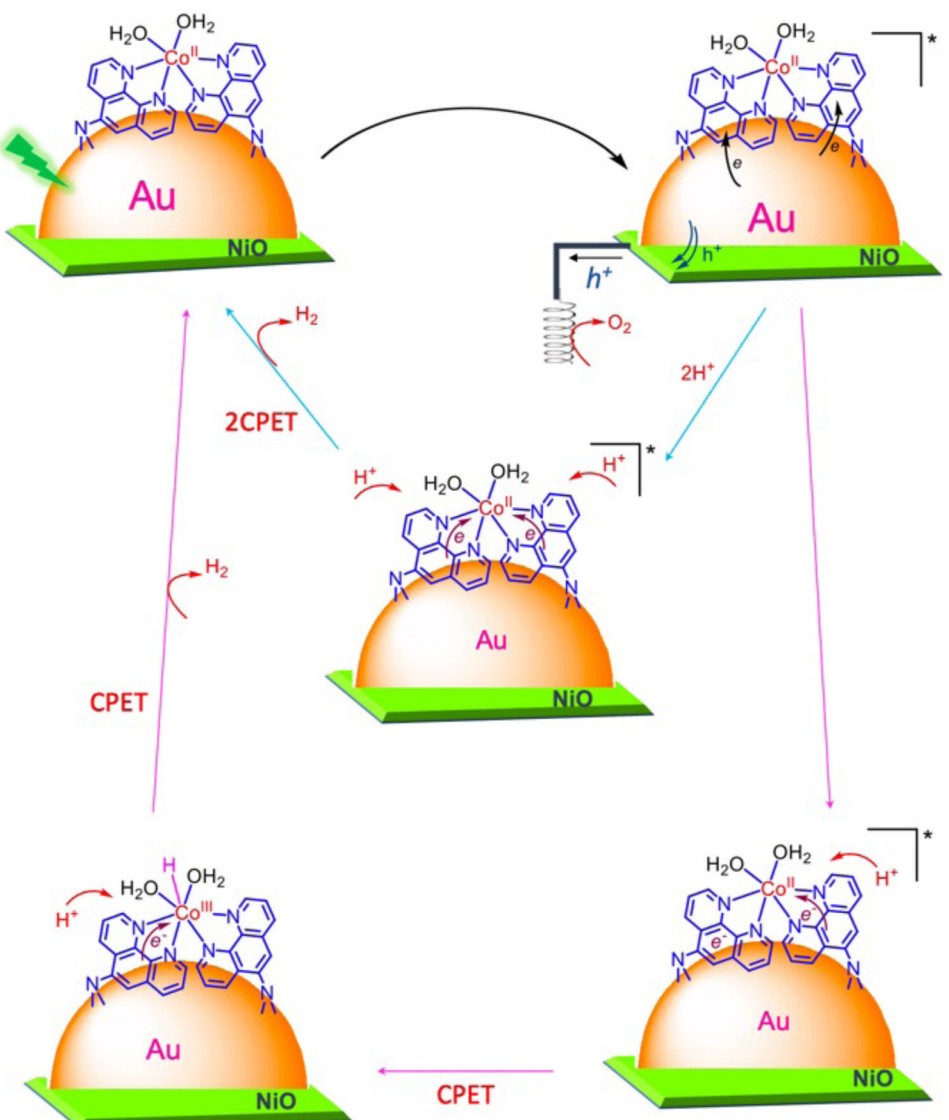

**Fig. 7 | Schematic representation of the catalytic cycle leading to H₂ evolution.** The figure shows two hypotheses consistent with the reported data: the simultaneous mechanism (*blue arrow pathway*) and the sequential mechanism (*pink arrow pathway*).

## Preparation of the thin films

**The preparation and process of NiO film.** The NiO paste was purchased from Solaronix (Ni-Nanooxide N/SP,~20 wt. %) and used as received. A small portion of the NiO paste was placed on a screen on top of the cleaned FTO glass and manually printed on the conducting side of the FTO glass. Then, the NiO-printed FTO glass plates were annealed at 500 °C for 1 h at a rate of 10 °C/min.

**Assembly of the Au nanoparticles on NiO film.** The synthesised Au nanoparticles were sprayed manually on NiO films and annealed at 500 °C for 1 h at a rate of 10 °C/min.

**Assembly of the catalyst on NiO/Au surface.** The annealed NiO-Au films were dipped into a 4 mg/mL water solution of the [$Co^{II}$(phen-$NH_2$)$_2$($H_2O$)$_2$] catalyst for 3 days. Finally, we could get the self-assembled NiO/Au/[$Co^{II}$(phen-$NH_2$)$_2$($H_2O$)$_2$] composite system. The system was rinsed with water several times to remove unbounded catalyst molecules. For experimental purposes, we have sprayed Au nanoparticles only on FTO glass and on FTO/NiO and attached the molecular linker using the same procedure to deposit the catalyst.

## Samples characterisation

**Inductively coupled plasma-optical emission spectrometry (ICP-OES) measurement.** The ICP-OES was used to estimate the cobalt amount in the sample. The film was digested in 4 mL of nitric acid (FisherScientific, Nitric acid 65%) for several hours. Before measurement, the small probe was diluted 10 times with Milli-Q water containing 2% $HNO_3$ and filtered with 0.2 μm syringe filters (Whatman). Avio 200 Scott/Cross-Flow Configuration was used for ICP measurements. A calibration curve was formed for the measurements using a Cobalt Calibration Standard (CPAchem). Concentrations of 0, 0.1, 1 and 10 ppm of the Co were used to create a 4-point linear regression. All measured values are within a relative standard deviation (RSD) of 2%.

**Electrochemistry measurements.** The electrochemical data were measured using an EmStat potentiostat instrument. For the electrochemical experiments, a typical cylindrical closed cell was used. The FTO films were placed on the side of the cell so that it could face the light. Acetic acid (Merck, ACS reagent ≥ 99.7%) with a 3 M concentration was used as a proton source.

**Bulk electrolysis.** In the presence of 0.1 M LiCl (Merck, ACS reagent ≥ 99%) as a supporting electrolyte and glassy carbon as a working electrode, a Pt wire as a counter electrode, Ag/AgCl (3 M KCl in water, Merck) as reference electrode was used for the bulk electrolysis experiment.

**Photo-electrocatalytic chronoamperometry.** The photoelectrode exposed area to light is 0.79 cm². Plasmonic excitation was performed with a 532 nm laser of 43.8 mW/cm² intensity. Pt wire counter electrode and Ag/AgCl (3 M KCl in water) reference electrode were purchased from Redox.me and used as received. Lithium chloride purchased from Merck was used as a supporting electrolyte without further purification. Acetic acid (Merck, ACS reagent ≥ 99%) was used as a proton source.

**Mass spectrometry analysis.** We measured the gas produced during the photoelectrochemical reaction using a quadrupole mass spectrometer (QMS) (HPR 20) from Hiden Analytical. Continuous argon gas flow (15 mL/min) through the cylindrical electrochemical cell during the measurement. Before applying the electrochemical potential, we saturated the electrochemical cell with argon to get a stable argon signal. Argon, $O_2$, and $H_2$ were measured continuously with the SEM detector throughout the measurement.

**Transient absorption spectroscopy (TAS).** A 40-fs pulsed laser with a 3 kHz repetition rate was generated through the Libra Ultrafast Amplifier System designed by Coherent. An optical parametric oscillator (TOPAS- prime, Light Conversion) created the excitation beam. The signals were detected with a UV-NIR detector from Newport MS260i spectrograph with interchangeable gratings. The fundamental laser (probe, 795 nm) passes through the delay stage (1–2 fs step size) and is focused in a Sapphire optical window to generate visible light from 400 to 750 nm. The instrument response function obtained for our system is ca. 95 fs.

**Transient infrared absorption spectroscopy (TIRAS).** A 40-fs pulsed laser with a 3 kHz repetition rate was generated through the Libra Ultrafast Amplifier System designed by Coherent. Two optical parametric oscillators (TOPAS- prime, Light Conversion) created the excitation beam and the probe light in the mid-IR (3000–10000 nm). The signals were detected with a Horiba iHR 320 spectrometer. The pump laser power was constantly monitored with less than a 2% standard deviation. The timing resolution, i.e., the instrument response function, is ca. 100 fs.

**Near-ambient pressure- X-ray photoelectron spectroscopy (NAP-XPS) measurements.** Details about the experimental chamber and the three-electrode setup have been described elsewhere[70]. NAP-XPS experiments were carried out at the PHOENIX I beamline of the Swiss Light Source Synchrotron (SLS), making use of the solid-liquid interface endstation in a three-electrode setup using a gold counter electrode and an Ag/AgCl reference electrode controlled via a potentiostat (BioLogic Science Instruments SP-300)[70]. Linearly polarised light was used throughout the experiments. The as-introduced sample was first analysed under high vacuum conditions to acquire reference spectra. Then, the chamber was opened, and the beaker containing the pre-deaerated electrolyte was introduced (see Fig. S1). The chamber was pumped down in a controlled way, using a needle valve, to avoid electrolyte spilling and favour the pressure equilibration (around 20 mbar). Measurements were carried out using an excitation energy of 5000 eV.

Deconvolution of the O 1s spectra was performed after removing a Shirley background. Gaussian and Voigt-shaped components, whose positions were set according to past literature reports, were used to obtain the best correlation with experimental data (see Fig. S15). Fitting parameters (peak positions, full width at half maximum –FWHM- and % of Lorentian-Gaussian) are summarised in Table S1.

## Data availability
The data related to the figures in the paper are provided as Excel files in Source data. Additional data supporting this study's findings are available from the corresponding author upon request. Source data are provided with this paper.

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

## Acknowledgements

the authors thank the Paul Scherrer Institute for providing access to the Phoenix beamline at the Swiss Light Source. The authors also thank Ms Johanna Andersson from Uppsala University for the lab access and her support with complex melting point measurements. The authors also thank Prof. Leif Hamarström from Uppsala University for the productive discussions about molecular catalyst reaction mechanisms and CPET aspects. Finally, the authors thank the MyFab clean room facility at the Angstrom Laboratory-Uppsala University for access to the clean room instrumentation. This work was funded by the Olle Engkvists stiftelse [210-0007 (J.S.)]; Knut & Alice Wallenberg Foundation [2019-0071(J.S.)]; Swedish Research Council [2019-03597 (J.S.)]; and Polish Ministry and Higher Education [1/SOL/2021/2 (A.W.)].

## Author contributions

A.D. and J.S. conceived the idea and designed experiments. A.D. and A.M. prepared the materials. A.D., A.M., R.B.V., V.R.S., A.W., P.M.L., Z.N., C.S., L.A. and J.S. performed the basic characterization, electrochemical studies, catalysis and advanced spectroscopies. A.D., R.B.V., V.R.S., A.W., P.M.L., Z.N., C.S., L.A. and J.S. analysed the experimental data and co-wrote the manuscript. All authors contributed to the discussions and manuscript preparation.

## Funding

## Competing interests

The authors declare no competing interests.
