## [Peer Review File · Nature Communications]

Hydrogen evolution with hot electrons on a plasmonic-molecular catalyst hybrid systemREVIEWER COMMENTS

Reviewer #1 (Remarks to the Author):

In this work, the authors have been developed a system consisting of NiO/Au/[CoII(phen-NH₂)₂(H₂O)₂] for photocatalytic hydrogen evolution. The reaction mechanism in this system could be explained by situ and ultrafast spectroscopic measurements. However, several issues still remain controversial and need to be solved. Therefore, I do not recommend the publication of this work.

Q1. It was claimed that the CoII(phen-NH₂)₂(H₂O)₂ was connected on the surface of AuNP. However, no evidence was provided. Could you provide some evidence? What is the interaction? A shift of XPS Au 4f spectra may help to reveal the interaction. In addition, the amino groups are difficult to be bounded on AuNP due to the steric effect.

Q2. The authors applied the composite structure of NiO/Au/[CoII(phen-NH₂)₂(H₂O)₂] for photocatalytic hydrogen evolution. However, no data on HER rate was provided. In addition, the HER rate of this reaction system needs to be compared with that reported in other photocatalytic systems.

Q3. This article repeatedly mentions in the abstract, conclusion, and introduction that this catalytic process is not related to local thermal effects. However, in Line 30-31, Page 6, from the results of photocurrent(Figure 2C), it is concluded that the results are related to heating effects from the illumination and plasmonic decay. This is clearly a conflict. Please give an explanation. On the other hand, an increase in temperature may definitely speed up the reaction based on basic chemistry.

Q4. As you said, Lu group reported a composite system of plasmonic AuNPs and molecular catalysts. In this report, both plasmonic hot carriers and localized thermal effects contribute to the efficient photocatalytic systems. Your statement on this report is not accurate. Please correct the expression in the text.

Q5. Can the authors provide morphology of the AuNPs sprayed onto NiO films? Previous literature has reported that the morphology of AuNPs could change significantly after high-temperature annealing (Langmuir, 2012, 28, 25, 9885.) So it is necessary to provide the morphology of the AuNPs before and after thermal annealing.

Q6. All NAP-XPS spectra need to be reanalyzed after peak deconvolution. The position of the binding energy of the element is accurate only when the peaks were properly deconvoluted.

Q7. Can the authors clarify the absence of the peak of XPS Co 2p? The peak shift of Co 2p was not evidenced by the results. Therefore, the mechanism in Figure 5 is based on guess not evidence.

Q8. The results in Figure 2a, claimed a stepwise reduction process. However, no further evidence was provided. In addition, the results can also be explained by electrochemical reduction of acetic acid.

Q9. It was claimed that the disappearance of the amino bonds in FTIR evidence the binding of amino group on AuNPs. This claim is not accurate. The disappearance of this peak can also be explained by the small amount of the molecules adsorbed on AuNPs. This explanation seems more possible. It is better to show the peaks of other vibrational bands from adsorbed molecules.

Q10. How the molecules were adsorbed on AuNPs? After the dipping, was the substrate rinsed with solvent to remove excess molecules?

Q11. In Figure 2a-b, the unit for x axis need to be clarified. Was this voltage referenced to Ag|AgCl or standard hydrogen potential?

Q12. On page 6, it was claimed that no hydrogen was detected in absence of molecular catalyst? However, no data was provided.

Q13. It was claimed that the XPS O 1s peak at 532.4 eV can be attributed to the liquid water from thin electrolyte layer. However, this may also, at least partially, come from the acetic acid added in the system.

Q14. The data of the Au catalyst in the control group is incomplete, so readers cannot clearly understand the comparison of photocatalytic performance of the newly synthesized catalyst. I advise an additional set of experiments of the effect of light in the chronoamperometry of the Au applying -0.65V potential in 3 mM acetic acid and 0.1M LiCl is suggested (pH = 3.5) is shown in Figure 2C.

Q15. There is a problem with the abbreviation of the phrase "proton-coupled electron transfers" (CEPT) in the text, which should be (PCET). This error also appears in Figure (5), which easily affects readers' reading, please note the correction.

Q16. The writing needs further polishing.

Reviewer #2 (Remarks to the Author):

This manuscript explores the combination of a plasmonic particle with a molecular complex for hydrogen generation under plasmon excitation. The system is nicely design and the authors used a very elegant combination of conventional characterization techniques with ultrafast and NAP-XPS, which are less standard in this type of studies. As such they proposed that plasmonic hot-carriers are involved in the process and they discarded thermal effects due to the low thermal stability of the molecular complex which seems not to degrade. Overall this is an interesting study, that besides the results, presents interesting combination of techniques to further debate the role of plasmons in enhanced catalytic processes. I think this manuscript could be of potential interest after sorting some key points:

Why do the authors neglect the effect of enhanced near-fields that can promote electronic excitations in the molecular complex (i.e. resembling homogeneous photocatalysis processes)? From the abstract, introduction and discussion the dispute is heat or hot-carriers but the enhanced fields are never discussed nor considered.

I am also wondering if (under a field-driven scenario), the e-ph coupling times could also be explained without needing the charge-transfer.

Also, it has been shown that the highly concentrated electric fields can change the water molecules adsorption orientation and reactivity, modifying reaction energy barriers. As such, the hypothesis that thermal water-splitting needs more than 500 degrees, I'm not sure if still holds under high e-fields conditions.

Figure 2 should have the illumination conditions.

The thermal argument is debatable. I agree with the authors, but it's also true that they should show (by external heating) the degradation of the molecular complex (to support the idea that under high temperatures the catalyst breaks down).

The NAP-XPS and TAS measurements are very nice and not usually found together in the same manuscript. So I thank the authors for trying to do a very deep mechanistic understanding with non-conventional (or not the most commonly used) techniques.

I think the scheme in figure 5 can be highly enhance. For instance, the authors could put the times measured (instead of "very fast"). We all know these processes are very fast. Saying only very fast is not new/relevant. Also, where are the holes in that scheme? I think omitting the NiO doesn't help to understand the mechanism. The first arrow is the plasmon excitation and decay (I guess, because it says nothing).

Reviewer #3 (Remarks to the Author):

The authors report a plasmonic photocatalytic system for plasmon-assisted electrochemical hydrogen evolution and use photoelectrochemistry, transient spectroscopy, and XPS to provide evidence for the electron and hole transfer from plasmonic Au nanoparticles to NiO hole acceptor and Co-catalyst as an electron acceptor. Using transient spectroscopies and XPS, the authors provide evidence for the charge transfer from Au nanoparticles to holes and electron acceptors. However, the following concerns remain unaddressed and I recommend this study to be published after major revision.

1) In the introduction, the authors state that "it remains challenging to disentangle charge carrier catalysis from photothermal effects". I agree with the authors here however, the same challenge still persists in the current study. Even though the authors have provided evidence for the charge transfer catalysis, authors have not provided any evidence to prove that photothermal effects are not playing a role in the photocatalytic system reported by the authors. The authors state that the photocatalytic system is not stable at which the water's thermolysis takes place, which is why photothermal catalysis may not be occurring in their experiments. Even a small increase in the temperature due to photothermal heating can decrease the activation energy and can catalyze the electrochemical hydrogen evolution. The temperature does not have to reach 500 – 2000C (as mentioned by the authors in the manuscript) for photothermal heating to catalyze electrochemical hydrogen evolution. Hence, due to the lack of evidence to prove that photothermal heating is not participating in the electrochemical HER in the photocatalytic system reported in this study, both charge transfer and heating are likely to be playing the role in the photocatalytic HER reported by the authors unlike claimed otherwise by the authors. Authors are requested to include relevant discussion.

2) To understand the relative contribution of the charge carriers and photothermal heating in catalyzing electrochemical HER, using a photocatalytic system involving non-plasmonic Au (smooth Au film) may help. NiO/smooth Au film/Co-Cat (non-plasmonic substrate) when used for the photocatalytic experiments, smooth Au being non-plasmonic, laser light excitation will mainly lead to photothermal heating/interband transition of Au and it would be possible to only understand the contribution of only photothermal heating and this seems to be the primary motivation behind the study.

3) Although the authors provide chronoamperometry data for the photoelectrocatalytic HER (Figure 2C) with light illumination on and off, CV data reporting electrocatalytic HER (with NiO/AuNPs/Co-Cat) with and without light illumination should be provided. Monitoring the onset potential for HER in conditions with lights on and off will provide further insights into the charge transfer process. Reporting the above-mentioned CV experiment with different light intensities will also provide additional evidence for the charge transfer.

4) Hot electron generation and transfer are dependent on the wavelength of the light. The authors have performed experiments only at a single wavelength of light. Control experiments reporting photoelectrochemistry and transient spectroscopies using light of wavelength which off-resonance (for example 642 nm which does not excite the plasmon resonance) of the Au particles should be reported. Comparing results obtained at 550 nm excitation (already included in the manuscript) with at least one off-resonance wavelength excitation can provide additional insights.

5) Ex-situ transient spectroscopies make it clear that including NiO in a photocatalytic system alters e-e scattering, and e-ph scattering lifetimes, however, For Figures 2B and 2D, data for only FTO/Au/Co-Cat (no NiO) should also be provided to prove that NiO is playing during photoelectrochemical HER.

6) Was the Argon atmosphere maintained during CV experiments reported in Figure 2B? If not, peak -0.5 V (present in both CVs NiO/Au/Co-Cat and NiO/Au), may also correspond to the oxygen reduction. Authors are requested to include the discussion regarding the same.

7) In Figure 2D, NiO/Au also produced photoelectrochemical current, rather in the opposite way. The authors are requested to include a discussion regarding this.

8) In the proposed mechanism section, authors state that the hot holes are transferred to the NiO and react at the counter electrode leading to O₂ production. Authors are requested to provide data for this statement and include relevant discussion.

9) Regarding the characterization of the catalytic system:

a) Only DLS measurements for the Au NPs are provided. DLS measurements are generally carried out in a colloidal state, however, in the photocatalytic system reported in this study, Au NPs are drop cast on NiO/FTO and then annealed. Drop casting and annealing of the drop casted Au NPs at 500°C will result in Au NPs aggregation which may shift the LSPR of the Au NPs. SEM images of Au NPs on FTO before and after annealing should be provided for thoroughness.

b) In Figure 2A, the authors state that no distinguishable peaks were reported before the catalytic wave which relates to Co-complex redox behavior. However, there is a clear reduction (-1.2 V in acid, -1.4 V in water) and oxidation peak in both CVs (water and acid) which are uncharacteristic of HER because in HER, oxidation peak on the reverse sweep is not usually observed (Figure 2C inset). Hence, out of two catalytic waves, the first catalytic wave is unlikely from HER and may be related to the Co-Cat oxidation-reduction behavior. Authors are requested to provide further discussion regarding this. Further, was the Co-cat free in the solution or deposited on the glassy carbon? Authors are requested to include this information in the SI.

c) Figure 3D reports the use of Au/ligand. No procedure for preparing Au/ligand (no Co) is reported in the paper. Authors are requested to provide the method of preparation in the SI.

d) Au nanoparticles are prepared using tannic acid. What is the purpose of using tannic acid as Au NPs of the same size can be prepared using citrate? Please include the relevant discussion for better clarity.

10) Regarding transient spectroscopy:

a) The lifetime of all the scattering processes changed after including Co-Cat in the photocatalytic system. Short pulses used for the spectroscopy may generate a lot of local heat which may damage the photocatalytic system especially the organic compound Co-complex. Can this decomposition or damage to the catalyst have an impact on the scattering process lifetime? Please include relevant discussion. FTIR, SEM, and UV Vis of the catalytic system after irradiating short laser pulses should be included to ensure the integrity of the catalyst and make sure that the decomposition of the catalyst (if at all) is not the reason behind the scattering process's lifetime change.

b) Why was the 490 nm winglet considered for the analysis and not the other winglet? Please provide the discussion.

11) It would be helpful for the readers if you can please provide pictures of the electrodes, electrochemical cell and experimental set up.

Answers to the Reviewers' comments

Reviewer #1:

In this work, the authors have been developed a system consisting of NiO/Au/[Coll(phen-NH₂)₂(H₂O)₂] for photocatalytic hydrogen evolution. The reaction mechanism in this system could be explained by situ and ultrafast spectroscopic measurements. However, several issues still remain controversial and need to be solved. Therefore, I do not recommend the publication of this work.

REPLY: We thank the Reviewer for the comprehensive revision and detailed feedback, which helped us improve significantly the quality of the manuscript. We have addressed all the comments below and changed the manuscript and SI accordingly. We also want to thank the Reviewer for the time dedicated to the manuscript revision. We hope the revised version suppressed the earlier reluctance and can be considered for publication.

Q1. It was claimed that the Coll(phen-NH₂)₂(H₂O)₂ was connected on the surface of AuNP. However, no evidence was provided. Could you provide some evidence? What is the interaction? A shift of XPS Au 4f spectra may help to reveal the interaction. In addition, the amino groups are difficult to be bounded on AuNP due to the steric effect.

REPLY: We thank the Reviewer for the suggestion. We used infrared to demonstrate that the catalyst was bounded to the Au. We see the disappearance of the NH₂ bands (Fig 1C insert) only when Au is present, corroborating our expectations. Additionally, the LSPR maximum shifted, suggesting that the surrounding dielectric medium changed when the catalyst was added. Finally, the XPS analysis requested by the Reviewer showed the Co signal is only present when Au is present, confirming anchoring selectivity.

The XPS analysis of the Au 4f revealed a doublet related to a single Au specie in all the samples (Au-Cat, NiO-Au, and NiO-Au-Cat). The Au 4f_{7/2} binding energy for all of them was found to be 84.05 eV, consistent with Au in metallic form. The single doublet suggests that Co 2p induced electronic changes are not localized, confirming good electronic coupling. Since the added catalyst was very small, the induced electronic changes couldn't be quantified.

Actions taken:

We updated the manuscript with the following statements:

The Au LSPR peak shifts to lower energy when the cobalt catalyst is added (Figure 1C), confirming the anchoring. Note that the LSPR peak absorption is sensitive to the surrounding dielectric medium. Consequently, the surface modification by the catalyst should induce a shift in the LSPR maximum as absorbed. Additionally, it is possible to see the complex absorption shoulder located at 370 nm, corroborating the attachment between catalyst and Au NPs. Unfortunately, the glass support (FTO or cover glass) covers the rest of the complex UV-Vis band precluding their measurement.

It is worth mentioning that the catalyst only bonded when Au NPs were present according to Co 2p XPS, confirming the selective anchoring of the catalyst to the Au NPs via the amino groups. The Co 2p_{3/2} of Au/Co-cat and NiO/Au/Co-cat measured in vacuum, at a prominent peak is centred at around 780.5 eV, consistent with Co is oxidation +2. The Au 4f_{7/2} for the samples Au/Co-cat., NiO/Au and NiO/Au/Co-cat. appeared at 84.0 eV (figure S9), consistent with gold in the metallic state. There was only one species in all the samples, coherent with the idea that any electronic change due to NiO and Co-catalyst is delocalized over all the gold atoms, suggest good electronic coupling.

The anchoring was also confirmed by the disappearance of the amino bands in the infrared (Figure 1C insert). Before anchoring the complex has three small peaks located between 3450-3300 cm⁻¹ and 3250-3200 cm⁻¹ associated with N-H bending modes of primary amino groups. The correspondent bending modes between 1650-1580 cm⁻¹ are also visible but somewhat overlapped by the water O-H bending mode. After attaching the catalyst to the Au NPs, the N-H bands disappear. The complete disappearance suggests that the catalyst coordinates to the Au NPs via both 1,10-Phenanthroline-5-amine ligands, as shown schematically in Figure 1D. Note that the infrared bands were normalized to the C-N stretch at 1280 cm⁻¹ intensity to enable

direct comparison. The C-N band is unaffected by the attachment, making it suitable for normalization. Unfortunately, the formed Au-N bonds are not infrared active, and the low loading prevented their detection with Raman spectroscopy. Figure 1D shows the a chematic representation of the complete photosystem.

- We added Figure S9, showing Au 4f XPS of Au/Co-cat., NiO/Au and NiO/Au/Co-cat. to the supporting information.

Q2. The authors applied the composite structure of NiO/Au/[CoII(phen-NH₂)₂(H₂O)₂] for photocatalytic hydrogen evolution. However, no data on HER rate was provided. In addition, the HER rate of this reaction system needs to be compared with that reported in other photocatalytic systems.

REPLY: We thank the Reviewer for the suggestion. In the original version of the manuscript, we had the photocurrent per area, which we confirmed by online QMS to be due to hydrogen production. We updated the manuscript with the rate (average of 3.1 nmol/(min*cm²)). We would like to reiterate that the aim of the manuscript is to demonstrate the direct use of hot carriers in the photocatalytic process. Moreover, because this is a new catalyst and its loading is purposely low, we think it is inappropriate to draw comparisons with unrelatable systems.

Q3. This article repeatedly mentions in the abstract, conclusion, and introduction that this catalytic process is not related to local thermal effects. However, in Line 30-31, Page 6, from the results of photocurrent(Figure 2C), it is concluded that the results are related to heating effects from the illumination and plasmonic decay. This is clearly a conflict. Please give an explanation. On the other hand, an increase in temperature may definitely speed up the reaction based on basic chemistry.

REPLY: We thank the Reviewer for the comment. What we wanted to say the drift in the baseline is related to heating effects, not the entire light-induced photocurrent signal. We updated the phrase to convey this. Heat cannot justify the increase in photocurrent upon illumination, since the magnitude of the baseline drift is much smaller than the photocurrent. This is a limitation of running experiments under longer illumination, which was mitigated by doing short on/off cycles (see Figure 2C). Nevertheless, we understand the Reviewer's natural confusion, and thus the reason for updating the text.

Q4. As you said, Lu group reported a composite system of plasmonic AuNPs and molecular catalysts. In this report, both plasmonic hot carriers and localized thermal effects contribute to the efficient photocatalytic systems. Your statement on this report is not accurate. Please correct the expression in the text.

REPLY: We thank the Reviewer for the comment.

We updated the manuscript with the following statement:

"Moreover, the authors suggested a cooperative result between plasmon hot carriers and localized thermal effects for which this study does not have evidence."

Q5. Can the authors provide morphology of the AuNPs sprayed onto NiO films? Previous literature has reported that the morphology of AuNPs could change significantly after high-temperature annealing (Langmuir, 2012, 28, 25, 9885.) So it is necessary to provide the morphology of the AuNPs before and after thermal annealing.

REPLY: We thank the Reviewer for the comment. Au NPs morphology after annealing is the same according to the AFM analysis (figure S6), which shows an average particle size in the range of 10 nm as expected from DLS analysis. Small particle aggregation cannot be discarded but is not the prevalent morphology. Regarding the NiO, the AFM shows that the deposition of Au and subsequent annealing did not affect its morphology (figure S7). Moreover, the Ni 2p XPS showed no significant change in the Ni binding energy to suggest that the addition of Au affected the NiO significantly from an electronic perspective (figure S8). This is not surprising since the supplier (Solaronix) stated that films can be annealed up to 700 C.

Actions taken:

- We added Figure S6, S7 and S8 to the supporting information.

Q6. All NAP-XPS spectra need to be reanalyzed after peak deconvolution. The position of the binding energy of the element is accurate only when the peaks were properly deconvoluted.

REPLY: We thank the reviewer for this comment. We agree that the shifts of the O 1s shown in the manuscript (Figures 4A and S8A, insets) should be supported by spectra deconvolutions. Indeed, in the first version of the manuscript, we slightly overestimated the shift of the main peak of O 1s. The correct shifts are now reported in the insets of Figure 4A and Figure S8A. Upon fitting of the O 1s, peak components' assignments were amended in the main text. We deconvoluted all spectra. As suggested by this reviewer, fitting results gave us precise peak positions and a correct estimate of the peak shift upon application of the voltage. In the revised version, we show deconvoluted spectra in Figure S15.

In the case of Co 2p, because (as written in the manuscript) the S/N is low and the spectral shape broad, a deconvolution would not be meaningful. Line shapes and peak positions are comparable to other reports of cobalt complexes, and this supports our qualitative description. Furthermore, Co 2p peaks position and shape do not change with the applied voltage. In the manuscript, we never report absolute binding energy values. We identify the binding energy of the main peak centroid.

Actions taken:

- We replaced the insets of Figures 4A in the manuscript

- We replaced the insets of Figures S14A in the supporting information

- We added Figure S15, showing O 1s peak deconvolutions, to the supporting information.

- We added the Table S1 in the methods section describing XPS measurements in the supporting information.

Position (eV)	FWHM	% L-G
529.50	1.75	0
531.00	1.85	25
532.68 – 532.75	1.75	0
534.80 – 535.25	0.90	0

We added to the supporting text:

“Deconvolution of the O 1s spectra was performed after removal of a Shirley background. Gaussian and Voigt-shaped components, whose position set according to past literature reports, were used to obtain the best correlation with experimental data (see Figure S15). Fitting parameters (peak positions, full width at half maximum –FWHM- and % of Lorentian-Gaussian) are summarized in Table S1.”

We modified the manuscript text with:

“Three prominent peaks, centered at approximately 531.00 eV, 532.70 eV and 534.90 eV, are assigned to adsorbed hydroxyls, liquid water (thin electrolyte film on top of the electrode) and gas phase water, respectively (fitting parameters are reported in Table S1). The slight shift of the O 1s main peak measured between pure electrolyte and the same in the presence of acid (from 532.68 to 532.75 eV under OCP conditions) may be due to the presence of acetic acid, whose contribution falls within the spectral range of liquid water peak. A fourth peak, centered at 529.50 eV, was detected in the absence of acid and assigned to lattice oxygen (electrode). Such a difference between the two conditions (no lattice oxygen detected in the presence of acid) suggests that the liquid electrolyte film in the absence of acid was thinner than in the presence of acid.”

Q7. Can the authors clarify the absence of the peak of XPS Co 2p? The peak shift of Co 2p was not evidenced by the results. Therefore, the mechanism in Figure 5 is based on guess not evidence.

Reply: we believe that this reviewer refers to the low intensity/noisy signal of Co 2p. Indeed, as reported in Figures 4B and S8B and mentioned in the text, such a signal has always been detected. The low intensity of Co 2p is due to photoelectrons attenuation through the liquid layer of electrolyte stabilized on top of the electrode. The spectra shown in Figures 4B and S8B

are the sum of more than 100 iterations, corresponding to several hours of acquisition. Due to limited beamtime duration, we had to stop measurements in order to acquire a complete dataset.

Concerning the mechanism, the reviewer is correct: no peak shift of Co 2p was detected upon voltage application by means of in situ XPS measurements. We justify this in the main text, on page 11: *“Figure 4B also shows that at -0.65 V vs Ag/AgCl the cobalt signal decreases in intensity. Such a potential is sufficient for the evolution of H₂ since bubbles were detected at the electrode. Thus, the decrease in signal suggests that some of the cobalt centres undergo fast reduction and protonation, leading to hydrogen evolution. However, since these are expected to be transient species, one cannot capture them due to our low signal-to-noise data and temporal resolution. Nevertheless, their presence can be deduced by the drop in the Co²⁺ signal intensity.”*

To better highlight the presence of Co 2p, and explain the low S/N ratio, **we modified the manuscript text with:**

“Despite the low signal-to-noise ratio, due to Co 2p attenuation through the liquid electrolyte layer stabilized on the WE, the main features of Co 2p peaks can be detected on both investigated electrodes (see figures 4B and S8B).”

Q8. The results in Figure 2a, claimed a stepwise reduction process. However, no further evidence was provided. In addition, the results can also be explained by electrochemical reduction of acetic acid.

REPLY: We thank the Reviewer for the comment. We remeasured several times figure 2a, ensuring that no oxygen was present in the solution. It became clear that peak was related to oxygen reduction and not catalyst reduction, which is consistent with NAP-XPS. The figure was updated accordingly.

Q9. It was claimed that the disappearance of the amino bonds in FTIR evidence the binding of amino group on AuNPs. This claim is not accurate. The disappearance of this peak can also be explained by the small amount of the molecules adsorbed on AuNPs. This explanation seems more possible. It is better to show the peaks of other vibrational bands from adsorbed molecules.

REPLY: We thank the Reviewer for the comment. The Reviewer is correct in suggesting that low loading of the catalyst could make detection of -NH₂ bands. To mitigate this limitation the FTIR signals were normalized to the C-N band at 1280 cm⁻¹, enabling direct comparison between unbound and bounded catalyst (see new figure 2C insert).

Upon binding, one expects the formation of Au-N bond, which is not active in the mid FTIR region. The bond is active in Raman but the low loading of catalyst made it impossible its detection. Unfortunately, the catalyst has no active FTIR bands that can be used to confirm the attachment.

Actions taken:

- We rechecked the FTIR and UV-Vis analyses and performed additional XPS to confirm the attachment. See the reply to Q1 for more information.

Q10. How the molecules were adsorbed on AuNPs? After the dipping, was the substrate rinsed with solvent to remove excess molecules?

REPLY: We thank the Reviewer for the comment. After attaching, the electrode was rinsed several times with the solvent to remove any excess of catalyst.

We updated the supporting information with the following statement:

“The system was rinsed with water several times to remove unbounded catalyst molecules.”

Q11. In Figure 2a-b, the unit for x axis need to be clarified. Was this voltage referenced to Ag|AgCl or standard hydrogen potential?

REPLY: We thank the Reviewer for the comment. The potential is versus Ag/AgCl as stated in the figure caption. However, we took the Reviewer suggestion and updated the figures to make this clear.

Q12. On page 6, it was claimed that no hydrogen was detected in absence of molecular catalyst? However, no data was provided.

REPLY: We thank the Reviewer for the comment. The QMS signal for the system without catalyst was the same as the H₂ baseline and thus we could confidently state that no hydrogen was produced. We opted against showing that because it is not common to show zero signals. However, to satisfy the Reviewer request we added the O₂ trace to Figure S4.

Q13. It was claimed that the XPS O 1s peak at 532.4 eV can be attributed to the liquid water from thin electrolyte layer. However, this may also, at least partially, come from the acetic acid added in the system.

REPLY: we thank the reviewer for the suggestion. We did not explicitly consider acetic acid in the discussion because its concentration, around 40 μM, is very low. In the revised version of the manuscript, we address this comment. We refer to Q6: *"The slight shift of the O 1s main peak measured between pure electrolyte and the same in the presence of acid (from 532.68 to 532.75 eV under OCP conditions) may be due to the presence of acetic acid, whose contribution falls within the spectral range of liquid water peak."* We added a reference that supports our claim (J. Phys. Chem. A 2013, 117, 401–409).

Q14. The data of the Au catalyst in the control group is incomplete, so readers cannot clearly understand the comparison of photocatalytic performance of the newly synthesized catalyst. I advise an additional set of experiments of the effect of light in the chronoamperometry of the Au applying -0.65V potential in 3 mM acetic acid and 0.1M LiCl is suggested (pH = 3.5) is shown in Figure 2C.

REPLY: We thank the Reviewer for the comment. We added the requested data to Figure 2D. It is visible that only when the complete system is present we get significant photocurrents.

Q15. There is a problem with the abbreviation of the phrase "proton-coupled electron transfers" (CEPT) in the text, which should be (PCET). This error also appears in Figure (5), which easily affects readers' reading, please note the correction.

REPLY: We thank the Reviewer for the comment. The Reviewer is correct if the abbreviation was for *proton-coupled electron transfer* but in the present case it is for *concerted proton-coupled electron transfer* that is commonly abbreviated as CEPT.

Q16. The writing needs further polishing.

REPLY: We thank the Reviewer for the comment. The manuscript was polished and checked by a third party.

Reviewer #2:

This manuscript explores the combination of a plasmonic particle with a molecular complex for hydrogen generation under plasmon excitation. The system is nicely design and the authors used a very elegant combination of conventional characterization techniques with ultrafast and NAP-XPS, which are less standard in this type of studies. As such they proposed that plasmonic hot-carriers are involved in the process and they discarded thermal effects due to the low thermal stability of the molecular complex which seems not to degrade. Overall this is an interesting study, that besides the results, presents interesting combination of techniques to further debate the role of plasmons in enhanced catalytic processes. I think this manuscript could be of potential interest after sorting some key points:

REPLY: We thank the Reviewer for recognition of the scientific merit of this work and recommendation. We also thank the Reviewer for the comprehensive revision and detailed feedback, which helped us improve significantly the quality of the manuscript. We have taken all the suggestions on board, and modify the manuscript accordingly. A detailed point-by-point clarification is presented below. We hope the revised manuscript matches the Reviewer's expectations and can be subsequently considered for publication.

Why do the authors neglect the effect of enhanced near-fields that can promote electronic excitations in the molecular complex (i.e. resembling homogeneous photocatalysis processes)? From the abstract, introduction and discussion the dispute is heat or hot-carriers but the enhanced fields are never discussed nor considered.

I am also wondering if (under a field-driven scenario), the e-ph coupling times could also be explained without needing the charge-transfer.

REPLY: We thank the Reviewer for the comment. We decided to answer both comments together since they are related. The Reviewer is correct in stating that effect of enhanced near-fields can also contribute to enhancements in catalysis.

We can consider two possibilities in the case of near-field enhancement:

- A. the plasmon enhances the optical absorption of the catalyst, similar to what Sheng et al. *Nat. Commun.* 14, 1528 (2023) published, or behaves as a strongly correlated system as proposed by Rossi et al. *Nat. Commun.* 10, 3336 (2019). In both cases, the catalyst and plasmon resonance optical absorptions should be close in energy to detect significant enhancements. As we showed our catalyst has an absorption in the UV region (ca. 370 nm), while of Au LSPR is centred around 550 nm, which is considerable far apart in energy to consider this kind of enhancement. Moreover, we performed the catalysis under monochromatic illumination with a 532 nm CW laser to ensure that only the plasmon is excited.
- B. Plasmon near-field induces an energy transfer to catalyst leading to increase in activity. The Reviewer is correct in stating that an energy transfer from plasmon to the catalyst would result in a decrease of e-ph lifetime, because lower energy in the resonance would induce faster relaxation of the system, similar to what happens when we transfer hot electrons. However, energy transfer cannot justify the TIRAS signal. The broad and featureless TIRAS signal that we detected when we have NiO and the catalyst, is indicative of free carriers that must have come from the plasmon as hot carriers.

While we cannot discard that near-field effects are present, the combined TAS and TIRAS measurements seems to be more supportive of hot carriers' involvement than energy transfer. Other effects such as increased optical absorption or establishment of a strong correlated systems are unlikely due to large energy difference between the LSPR and catalysts absorption. Finally, catalysis promotion by local electric field enhancement is possible but one still needs the hot electrons to reduce the protons.

We added to the manuscript text:

"The free carrier signal detected by TIRAS confirms the transference of hot carriers to the acceptors, suggesting that the observed decrease in e-ph lifetime is less likely to be due to energy transfer. Since the Au plasmon resonance absorption is very far from the acceptors' absorptions, one can also discard the hypothesis of photonic enhancement as the corporate for catalytic performance."

Also, it has been shown that the highly concentrated electric fields can change the water molecules adsorption orientation and reactivity, modifying reaction energy barriers. As such, the hypothesis that thermal water-splitting needs more than 500 degrees, I'm not sure if still holds under high e-fields conditions.

REPLY: We thank the Reviewer for the comment. Water molecules has strong bonds and consequently one needs high temperature to break them. In the case of water thermolysis in absence of a catalyst, this can be between 800-1000 C. Electric fields induce water molecules dipole alignment facilitating adsorption and increasing coverage, however the most significant effect seems to be on the electrical charges, in which electric fields can mitigate to some extent electron-hole recombination, and enhance the interaction of holes with water molecules at the interface, leading to more facile water intramolecular dissociation (Boyd et al *Energies* 15, 1553 (2022) and references within).

The statement suggest once more a promoter role for the electric field but the reaction still needs the carriers. Additionally, the enhancements are small and thus it is not expected a significant decrease in water thermolysis temperature. The reported 500 C value is a conservative estimate, measured in a presence of a catalyst.

We added to the text:

"There is the possibility for near-field enhancements caused by the local electric fields formed upon Au LSPR excitation. While localized electric fields can promote the catalytic process by improving charge separation and molecule dipole alignment, they act on the hot carriers. Consequently, the local electric fields cannot catalyze the process, i.e., one still needs electrons to reduce the protons that are not generated from the localized electric fields."

And:

The complete system's susceptibility to off-resonant excitation was also evaluated. Excitation at 650 nm (off-resonant) did not yield significant differences in the CV compared with no excitation (Figure S17). Off-resonance excitation creates local effects like an increased temperature (hot spots) and a high electric field. However, when it comes to hot carriers, their energy is low and consequently not valuable to drive photocatalytic processes. The findings suggest that hot electrons are involved in the catalysis, and the increase in catalytic output under illumination is related to hot electrons, not thermal or electric fields. However, this does not mean that they are not present and might help the catalysis; instead, they cannot justify the catalytic performance on their own.

Figure 2 should have the illumination conditions.

REPLY: We thank the Reviewer for the comment. We updated the manuscript accordingly.

The thermal argument is debatable. I agree with the authors, but it's also true that they should show (by external heating) the degradation of the molecular complex (to support the idea that under high temperatures the catalyst breaks down).

REPLY: We thank the Reviewer for the comment. The catalysts complex was found to thermally decompose at 265° C, well below water thermolysis.

We added to the text:

The molecular catalyst thermally decomposes at 265 °C.

The NAP-XPS and TAS measurements are very nice and not usually found together in the same manuscript. So I thank the authors for trying to do a very deep mechanistic understanding with non-conventional (or not the most commonly used) techniques.

REPLY: We thank the Reviewer for recognizing this. Combining NAP-XPS with ultrafast is indeed uncommon, and if we must say after doing it, we know why. The NAP-XPS measurements on a mesoporous electrode proved to be extremely challenging and according to the beam line responsible unreported till now. Mesoporous films capture too much electrolyte, requiring several hours removal until an unbroken thin film of electrolyte is established enabling XPS measurement under bias.

I think the scheme in figure 5 can be highly enhance. For instance, the authors could put the times measured (instead of "very fast"). We all know these processes are very fast. Saying only very fast is not new/relevant. Also, where are the holes in that scheme? I think omitting the NiO doesn't help to understand the mechanism. The first arrow is the plasmon excitation and decay (I guess, because it says nothing).

REPLY: We thank the Reviewer for the comment. We updated the figure to accommodate the Reviewer's suggestions.

Reviewer #3:

The authors report a plasmonic photocatalytic system for plasmon-assisted electrochemical hydrogen evolution and use photoelectrochemistry, transient spectroscopy, and XPS to provide evidence for the electron and hole transfer from plasmonic Au nanoparticles to NiO hole acceptor and Co-catalyst as an electron acceptor. Using transient spectroscopies and XPS, the authors provide evidence for the charge transfer from Au nanoparticles to holes and electron acceptors. However, the following concerns remain unaddressed and I recommend this study to be published after major revision.

REPLY: We thank the Reviewer for recognition of the scientific merit of this work and recommendation. We also thank the Reviewer for the comprehensive revision and detailed feedback, which helped us improve significantly the quality of the manuscript. We have taken all the suggestions on board, and modify the manuscript accordingly. A detailed point-by-point clarification is presented below. We hope the revised manuscript matches the Reviewer's expectations and can be subsequently considered for publication.

1) In the introduction, the authors state that "it remains challenging to disentangle charge carrier catalysis from photothermal effects". I agree with the authors here however, the same challenge still persists in the current study. Even though the authors have provided evidence for the charge transfer catalysis, authors have not provided any evidence to prove that photothermal effects are not playing a role in the photocatalytic system reported by the authors. The authors state that the photocatalytic system is not stable at which the water's thermolysis takes place, which is why photothermal catalysis may not be occurring in their experiments. Even a small increase in the temperature due to photothermal heating can decrease the activation energy and can catalyze the electrochemical hydrogen evolution. The temperature does not have to reach 500 – 2000C (as mentioned by the authors in the manuscript) for photothermal heating to catalyze electrochemical hydrogen evolution. Hence, due to the lack of evidence to prove that photothermal heating is not participating in the electrochemical HER in the photocatalytic system reported in this study, both charge transfer and heating are likely to be playing the role in the photocatalytic HER reported by the authors unlike claimed otherwise by the authors. Authors are requested to include relevant discussion.

REPLY: We thank the Reviewer for the comment. We want to start by emphasizing that we do not argue against some positive effects of the thermal process. We state that the photothermal process cannot alone justify the observed catalytic performance. This becomes even more obvious from the catalysis experiments performed under different light wavelengths, which the Reviewer suggested and will be discussed in point 4. We think the additional data, some of which was requested by the Reviewer, strengthens our conclusions and hopefully suppresses the Reviewer's original concerns.

2) To understand the relative contribution of the charge carriers and photothermal heating in catalyzing electrochemical HER, using a photocatalytic system involving non-plasmonic Au (smooth Au film) may help. NiO/smooth Au film/Co-Cat (non-plasmonic substrate) when used for the photocatalytic experiments, smooth Au being non-plasmonic, laser light excitation will mainly lead to photothermal heating/interband transition of Au and it would possible to only understand the contribution of only photothermal heating and this seems to be the primary motivation behind the study.

REPLY: We thank the Reviewer for the comment. The Reviewer's suggestion is excellent and was something we have tried. However, attaching the catalyst to any gold film was impossible despite many attempts. Au films prepared by evaporation or electrodeposition showed no signature for catalyst attachment in the CVs, precluding us from making the desired experiment. We suspect the catalyst is attached to unsaturated Au atoms exposed by the annealing and consequent removal of the citrate capping agent. This is why the loading on Au NPs is very low, and the catalyst doesn't attach to Au films. However, this requires further investigation that is beyond the work scope.

3) Although the authors provide chronoamperometry data for the photoelectrocatalytic HER (Figure 2C) with light illumination on and off, CV data reporting electrocatalytic HER (with NiO/AuNPs/Co-Cat) with and without light illumination should be provided. Monitoring the onset potential for HER in conditions with lights on and off will provide further insights into the charge transfer process. Reporting the above-mentioned CV experiment with different light intensities will also provide additional evidence for the charge transfer.

REPLY: We thank the Reviewer for the comment. The requested data was added to supporting information. We think the new figure shows the effect of light that the Reviewer requested. Since the data was collected with the same electrode, the measurements are comparable and quantitative.

Actions taken:

- We added Figure S17 to the supporting information.

4) Hot electron generation and transfer are dependent on the wavelength of the light. The authors have performed experiments only at a single wavelength of light. Control experiments reporting photoelectrochemistry and transient spectroscopies using light of wavelength which off-resonance (for example 642 nm which does not excite the plasmon resonance) of the Au particles should be reported. Comparing results obtained at 550 nm excitation (already included in the manuscript) with at least one off-resonance wavelength excitation can provide additional insights.

REPLY: We thank the Reviewer for the comment. The requested data was added to supporting information. The requested data was added to supporting information. We think the new figure shows the importance of exciting at the LSPR resonance maximum that the Reviewer requested. Since the data was collected with the same electrode, the measurements are comparable and quantitative. Moreover, this provides evidence that thermal cannot be the sole culprit for the catalysis since this would yield similar response independent of the excitation wavelength.

Actions taken:

- We added Figure S17 and S18 to the supporting information.

5) Ex-situ transient spectroscopies make it clear that including NiO in a photocatalytic system alters e-e scattering, and e-ph scattering lifetimes, however, For Figures 2B and 2D, data for only FTO/Au/Co-Cat (no NiO) should also be provided to prove that NiO is playing during photoelectrochemical HER.

REPLY: We thank the Reviewer for the comment. We updated the figures 2B and 2D to include with the requested data.

6) Was the Argon atmosphere maintained during CV experiments reported in Figure 2B? If not, peak -0.5 V (present in both CVs NiO/Au/Co-Cat and NiO/Au), may also correspond to the oxygen reduction. Authors are requested to include the discussion regarding the same.

REPLY: We thank the Reviewer for the comment. The peak at -0.5V is related to oxygen on Au reduction since it is always present when Au is in the system.

7) In Figure 2D, NiO/Au also produced photoelectrochemical current, rather in the opposite way. The authors are requested to include a discussion regarding this.

REPLY: We thank the Reviewer for the comment. We want to start by stating that photocurrents measured with NiO/Au are very small and, thus, challenging to conclude from shape analysis. However, since the QMS detected no H₂ from this system and NiO is a known OER catalyst, the current may originate from OER reaction on NiO surface. The amount is meagre and thus not detectable by the QMS but would induce a change in photocurrent direction. We want to emphasize again that the photocurrent is very low, and no products were detected in the QMS, so shape analysis must be done with extreme care. If what is happening

is what we stated, it further confirms the importance of plasmonic hot carriers to the overall process.

8) In the proposed mechanism section, authors state that the hot holes are transferred to the NiO and react at the counter electrode leading to O₂ production. Authors are requested to provide data for this statement and include relevant discussion.

REPLY: We thank the Reviewer for the comment. We overlapped to the SI to figure S4 the O₂ trace confirming that O₂ is produced at the counter electrode. The H₂:O₂ is roughly 2:1 as expected from the process stoichiometry.

9) Regarding the characterization of the catalytic system:

a) Only DLS measurements for the Au NPs are provided. DLS measurements are generally carried out in a colloidal state, however, in the photocatalytic system reported in this study, Au NPs are drop cast on NiO/FTO and then annealed. Drop casting and annealing of the drop casted Au NPs at 500°C will result in Au NPs aggregation which may shift the LSPR of the Au NPs. SEM images of Au NPs on FTO before and after annealing should be provided for thoroughness.

REPLY: We thank the Reviewer for the comment. We updated the SI with AFM images of Au nanoparticles after annealing at 500°C showing little to no aggregation even when measurements are carried out in a flat Si wafer. We also added AFM of the sample after second annealing showing that NiO morphology including porous structure is preserved. Finally, we did additional XPS analysis confirming that NiO surface oxidation is preserved throughout the preparation of the film.

b) In Figure 2A, the authors state that no distinguishable peaks were reported before the catalytic wave which relates to Co-complex redox behavior. However, there is a clear reduction (-1.2 V in acid, -1.4 V in water) and oxidation peak in both CVs (water and acid) which are uncharacteristic of HER because in HER, oxidation peak on the reverse sweep is not usually observed (Figure 2C inset). Hence, out of two catalytic waves, the first catalytic wave is unlikely from HER and may be related to the Co-Cat oxidation-reduction behavior. Authors are requested to provide further discussion regarding this. Further, was the Co-cat free in the solution or deposited on the glassy carbon? Authors are requested to include this information in the SI.

REPLY: We thank the Reviewer for the comment. We remeasured several times figure 2a, ensuring that no oxygen was present in the solution. It became clear that peak was related to oxygen reduction and not catalyst reduction, which is consistent with NAP-XPS. The figure was updated accordingly.

Actions taken:

- We added Figure S6, S7 and S8 to the supporting information.

c) Figure 3D reports the use of Au/ligand. No procedure for preparing Au/ligand (no Co) is reported in the paper. Authors are requested to provide the method of preparation in the SI.

REPLY: We thank the Reviewer for the comment. We added the description to the supporting information. This was done in the same manner as adding the catalyst by simple immersion in a solution containing the linker.

d) Au nanoparticles are prepared using tannic acid. What is the purpose of using tannic acid as Au NPs of the same size can be prepared using citrate? Please include the relevant discussion for better clarity.

REPLY: We thank the Reviewer for the comment. We followed Piella et al. *Chem. Mater.* 28, 1066 (2016) synthesis protocol. The protocol is very reproducible and reliable, and became the cornerstone of our group plasmonic synthesis. According to the authors and some studies performed in our group, tannic acid decreases the synthesis pH from 7.7 to 6.4. This promotes formation of smaller particles and a higher uniformity in size and shape. There are ample

literature that advocates pH role in controlling size and uniformity of particles produced with the Turkevich method.

10) Regarding transient spectroscopy:

a) The lifetime of all the scattering processes changed after including Co-Cat in the photocatalytic system. Short pulses used for the spectroscopy may generate a lot of local heat which may damage the photocatalytic system especially the organic compound Co-complex. Can this decomposition or damage to the catalyst have an impact on the scattering process lifetime? Please include relevant discussion. FTIR, SEM, and UV Vis of the catalytic system after irradiating short laser pulses should be included to ensure the integrity of the catalyst and make sure that the decomposition of the catalyst (if at all) is not the reason behind the scattering process's lifetime change.

REPLY: We thank the Reviewer for the comment. Beam damage is an issue that, unfortunately, has plagued several published data sets. Experienced users, which we considered to be, take the beam damage problem very seriously. We adopt several procedures that enable us to ensure that the data is collected unaffected by beam damage, such as:

- Sample circulation so the same spot is not measured more than 2 or 3 times
- Comparison of between spectra measured on the same spot
- Power dependence measurements to determine damage power threshold

By adopting such practices, one can ensure data without beam damage effects. Unfortunately, UV-Vis and FTIR analysis offer low guarantees because the spot size of the laser system is significantly smaller than the probe area of such analytics, meaning damage is only observed in extreme cases. SEM could detect pinhole formation due to pump laser damage; however, we are using fluencies where this is impossible, at least, from our experience

b) Why was the 490 nm winglet considered for the analysis and not the other winglet? Please provide the discussion.

REPLY: We thank the Reviewer for the comment. Similar analysis was performed with the other winglet, which yielded the same result.

We added to the manuscript text:

"Note that similar findings were obtained when performing the analysis on the winglet to the red of LSPR maximum"

11) It would be helpful for the readers if you can please provide pictures of the electrodes, electrochemical cell and experimental set up.

REPLY: We thank the Reviewer for the comment. We added some pictures of electrodes, electrochemical cell and experimental setup to the revised SI.

Actions taken:

- We added Figures S1 and S2 to the supporting information.

REVIEWER COMMENTS

Reviewer #1 (Remarks to the Author):

Although readers made some revision to improve the quality of the manuscript, there are still some issue to be resolved.

1. In response to Q1 from Reviewer 1, author tried to convince the successful binding of $\text{Co}(\text{phen-NH}_2)_2(\text{H}_2\text{O})_2$ via Au-N bonds. However, I am still not convinced. The IR spectra focused only the wavenumber of 3200-3500 cm^{-1} . It is strongly suggested to show the spectra of other wavenumber region for cross checking. Authors claimed that the binding of molecules was confirmed by the shift in UV-Vis spectra. However, the shift of UV-Vis peak is attributed to the change in refractive index of surrounding dielectric medium. This can also happen in the case of physical adsorption. Therefore, the chemical bonding of Au-N could not be confirmed by UV-Vis spectra. In term of XPS spectra, a shift of Au 4f and N 1s should be presence because of the Au-N bonding. However, no shift was observed. Therefore, the Au-N bonding could not be confirmed.
2. In response to Q2 from Reviewer 1, authors refused to compare their catalyst with similar one in the literatures by claiming their catalyst is new. It will be great to compare new catalyst with existing ones, which will make readers to understand the efficiency of the newly developed catalyst.
3. In response to Q6 from Review 1, after deconvolution, the shift of O 1s peak was only 0.15/0.1 eV, which is usually too small for a reliable shift. Usually, the shift smaller than 0.2 might be attributed to measurement error.
4. In response to Q15 from Review 1, it is still not logical to abbreviate "concerted proton-coupled electron transfers" as CEPT.
5. Since the claimed Au-N bonding was not convinced and the mechanism was not fully confirmed by results, I think the quality of this work is not high enough for being published in Nature Communications.

Reviewer #2 (Remarks to the Author):

The revised version of the manuscript is highly improved but some answers are still weak, in my opinion. There are some points that I still disagree; I mention them below. Overall, this is a very nice work and as I mentioned earlier in the first round, I thank the authors for putting together a very nice set of techniques that have rarely been used in plasmonic catalysis until now.

- 1) The explanation that the catalyst absorbs in the UV and that the plasmon resonance is in the visible is not enough - in my opinion - to discard the role of the electric field. There are many examples in the literature using catalytic metals (absorbing in the UV) but operating nicely in the visible when coupled to the plasmonic near-field of a secondary metal. Very similar to this case.
- 2) The thermolysis of the bounded catalyst should be shown by external heating. What do you detect

when the catalyst breaks down? This is the main support for discarding thermal effects and it should be better demonstrated.

3) I disagree with the discussion regarding the wavelength-dependent experiments (S17 and S18) and the fact that they further support a hot-carriers pathway. Exciting the plasmon resonance is also the most efficient way to heat the system, so both processes, photothermal heating (that depends on the absorption cross-section) and hot-carriers generation are maximized at the same wavelength. For that reason, I think that point 2 is relevant (to show the thermolysis by external heating more than wavelength-dependence).

As one of the key motivations of the paper is to disentangle the mechanisms behind the reported activity under plasmon excitation, I think it would be necessary to show in a more comprehensive way the role of field enhancement and heat.

Point-by-point answer to Reviewers comments:

Reviewer #1:

Although readers made some revision to improve the quality of the manuscript, there are still some issue to be resolved.

We thank the Reviewer for acknowledging the enhancements made to the manuscript and providing valuable feedback to strengthen its quality. Additional experiments were conducted to substantiate our initial claims about catalyst anchoring. Plus, we added some comparisons with catalysts that share similarities to ours regarding their bulk electrolysis. In our view, these were the main aspects highlighted. We appreciate the Reviewer's constructive feedback, suggestions, and the time invested in revising the manuscript. We trust that the revised version addresses any lingering concerns, paving the way for the acceptance of the manuscript.

1. In response to Q1 from Reviewer 1, author tried to convince the successful binding of CoII(phen-NH₂)₂(H₂O)₂ via Au-N bonds. However, I am still not convinced. The IR spectra focused only the wavenumber of 3200-3500 cm⁻¹. It is strongly suggested to show the spectra of other wavenumber region for cross checking. Authors claimed that the binding of molecules was confirmed by the shift in UV-Vis spectra. However, the shift of UV-Vis peak is attributed to the change in refractive index of surrounding dielectric medium. This can also happen in the case of physical adsorption. Therefore, the chemical bonding of Au-N could not be confirmed by UV-Vis spectra. In term of XPS spectra, a shift of Au 4f and N 1s should be presence because of the Au-N bonding. However, no shift was observed. Therefore, the Au-N bonding could not be confirmed.

We thank the Reviewer for the suggestion. We performed the suggested XPS experiments to substantiate the catalyst anchoring to the Au surface. The XPS comparing the N 1s and Au 4f signals before and after binding to the Au surface are presented in Figures 2E and 2F, respectively. Before attaching, the catalyst has two N1s peaks: the N from the phenanthroline bonded to the Co at 398.8 eV and N at 401.4 eV ascribed to NH₂ groups. Upon attaching, the signal related to N of the NH₂ group disappeared, and the N from phenanthroline shifted to 399.1 eV and got broader (FWHM before 1.641 and after attaching 1.812). These observations are consistent with attachment via the NH₂ groups and formation of Au-N species, which cannot be distinguished from the N of the phenanthroline ligand because of signal-to-noise. Additionally, there were no changes in the Au 4f, which is understandable since the catalyst loading was very low.

Action taken: we added Figure 2A and 2B to the manuscript and added to the manuscript:

The 10-Phenanthroline-5-amine ligand was purposely chosen to ensure selective coordination to the gold surface via the amino groups. This first supporting evidence came from XPS measured at low vacuum conditions, which had a Co 2p signal related to the catalyst only when Au NPs were present. The Co 2p_{3/2} on Au/Co-cat and NiO/Au/Co-cat measured had single contribution centred at around 780.5 eV, consistent with Co is oxidation +2. The observation that Co signal was only present when Au is present is a strong endorsement to the selective anchoring of the catalyst to the gold surface. The anchoring is believed to occur via the -NH₂ groups. This was corroborated by the disappearance of the amino bands in the infrared...

The UHV-XPS comparing the N 1s and Au 4f signals before and after anchoring the catalyst to the Au surface are presented in Figures 2A and 2B, respectively. Before attaching, the catalyst has two N 1s peaks: the N from the phenanthroline bonded to the cobalt center at 398.8 eV and N at 401.4 eV ascribed to the -NH₂ groups. Note that the UHV-XPS also did not show a peak ascribed to the nitrate

groups, corroborating the its exchange by water molecules. Upon attaching, the N 1s signal related to the -NH₂ group disappeared, and the N from phenanthroline shifted to 399.1 eV and got broader (FWHM before 1.641 and after attaching 1.812). Note that one expects the N 1s from the amino group to shift to lower binding energies as it loses the protons at a rate of about 1eV per each hydrogen atom lost. These observations are consistent with attachment via the NH₂ groups and formation of Au-N species, which cannot be distinguished from the N of the phenanthroline ligand because of signal-to-noise. The Au 4f_{7/2} had a binding energy at 83.6 eV for the sample with and without catalyst, consistent with metallic gold. There was only one species in all the samples, coherent with the idea that any electronic change due to NiO and Co-catalyst is delocalized over all the gold atoms, suggest good electronic coupling. Figure 1D shows the a schematic representation of the complete photosystem.

2. In response to Q2 from Reviewer 1, authors refused to compare their catalyst with similar one in the literatures by claiming their catalyst is new. It will be great to compare new catalyst with existing ones, which will make readers to understand the efficiency of the newly developed catalyst.

We thank the Reviewer for the comment. We did not refuse to compare the catalyst performance with other systems because we wanted to ensure that the comparison is fair and valid. In the revised version of the manuscript we compared our catalyst with Luo's et al system that served as inspiration for the reported system. In the new version, we compared our system further with Wang's et al. system, namely [Co(Py3Me-Bpy)(OH₂)]⁻ (PF₆)₂. Their proposed reaction mechanism is very different from ours (as with the case of Luo's bipyridine system) but the system has similar structural rigidity and coordination making it possible to compare.

Action taken: we added to the revised text:

Bulk electrolysis of the cobalt catalyst in water using 0.1 M LiCl as a supporting electrolyte without acid (Figure 3A) shows no unique reduction peaks before the onset of the catalytic wave at -1.18 vs Ag/Ag/Cl. Initiating with the absence of reduction peaks, this discovery diverges from the observations made by Luo et al. (whose system served as inspiration for this study) and the system studied by Wang et al., exhibiting comparable rigidity and coordination. The onset potential for electrolytic hydrogen production without using acid in our catalyst was lower, differing by only ca. 15 mV from Wang et al. system but by a noteworthy 190 mV from the Luo et al. system.

Predictably, the catalytic wave's amplitude is heightened with the presence of protons, underscoring proton availability as a pivotal factor in accessing catalytic performance within this system.

Consequently, all catalytic data was obtained in a 3 mM acetic acid environment. Although the introduction of acid did not result in new reduction peaks, it did cause a notable shift in the onset potential, reducing it by as much as 120 mV—a deviation from the characteristics of the previously documented system. These observations imply the engagement of two concerted proton-electron transfer mechanisms as opposed to the conventional sequential reduction followed by protonation. Subsequent sections will delve into additional evidence supporting this proposed mechanism.

3. In response to Q6 from Review 1, after deconvolution, the shift of O 1s peak was only 0.15/0.1 eV, which is usually too small for a reliable shift. Usually, the shift smaller than 0.2 might be attributed to measurement error.

We thank the Reviewer for the comment. We understand the Reviewer's uncertainty about our in situ XPS results, but we respectfully disagree with the comment. The experiment is complex and final shifts detected are small. However, we capitalized on several years of experience with such measurements and we believe that the shifts shown are real for two reasons:

1. Statistics of the measurement. Spectra shown in Figures 4 (main text), S14 and S15 do not display

single scans of the O 1s. They are the sum of 30 to 50 iterations. The signal-to-noise ratio is high and deconvolutions give good correlation with experimental data.

2. Deconvolutions of O 1s (Figure S15) show that peaks at 529.5 eV and 531.0 eV, corresponding to lattice oxygen and adsorbed hydroxyls, respectively, do not shift upon voltage application. In contrast, the component centered at approximately 532.7 eV, assigned to the thin electrolyte film on top of the electrode, displays a clear shift (spectra never superimpose in that region). If we consider lattice oxygen and hydroxyls as “internal references”, this is a clear evidence of voltage-induced shift of the 532.7 eV peak.

Actions taken: we modified the text as follows.

A difference of about 0.1-0.15 V was detected between the applied potential chosen based on the catalysis and seen in the O1s NAP-XPS, which is assigned to calibration shifts of the reference electrode during long experimental times. Therefore, the values specified in the plots are applied for consistency, not measured voltages. It is important to highlight that while the 532.7 eV peak component clearly displays a shift proportional to the applied voltage, components at 529.5 eV and 531.0 eV do not.

4. In response to Q15 from Review 1, it is still not logical to abbreviate “concerted proton-coupled electron transfers” as CEPT.

We thank the Reviewer for the comment. The Reviewer is correct in stating that it is not concerted proton-coupled electron transfer. Instead, it should be concerted proton-electron transfers (CPET) according to for example Tyburski et al. *J. Am. Chem. Soc.* 143 (2021) 560, which we corrected. This is indeed the terminology used by the leading groups

Action taken: we fixed the issue in the text and reaction mechanism figure (figure 7) by making sure that is concerted proton-electron transfers (CPET).

5. Since the claimed Au-N bonding was not convinced and the mechanism was not fully confirmed by results, I think the quality of this work is not high enough for being published in Nature Communications.

We thank the Reviewer for the comment. We are convinced that the newly added XPS and anchoring followed by in situ FTIR substantiates our claims and suppresses the original Reviewer’s reservations.

Reviewer #2:

The revised version of the manuscript is highly improved but some answers are still weak, in my opinion. There are some points that I still disagree; I mention them below. Overall, this is a very nice work and as I mentioned earlier in the first round, I thank the authors for putting together a very nice set of techniques that have rarely been used in plasmonic catalysis until now.

We thank the Reviewer for acknowledging the enhancements made to the manuscript and providing valuable feedback to strengthen its quality. Additional experiments were conducted to substantiate the direct involvement of hot carriers in the process. More specifically, we performed other chronoamperometry data under light modulation and analyzed the response shape, which provides additional evidence for the involvement of hot electrons in the catalytic process. We appreciate the Reviewer's constructive feedback, suggestions, and the time invested in revising the manuscript. We trust that the revised version addresses any lingering concerns, paving the way for the acceptance of the manuscript.

1) The explanation that the catalyst absorbs in the UV and that the plasmon resonance is in the visible is not enough - in my opinion - to discard the role of the electric field. There are many examples in the literature using catalytic metals (absorbing in the UV) but operating nicely in the visible when coupled to the plasmonic near-field of a secondary metal. Very similar to this case.

We thank the Reviewer for the comment. Significant enhancements due to near-field in plasmon-molecule hybrid systems were, to our knowledge, reported only when you have some optical overlap. For systems without optical overlap, those effects were only observed when you have the plasmonic materials in a core-shell architecture (Kholmicheva et al. ACS Nano 2014; 8, 12549 & Nanophotonics 2019; 8, 613), which is not what we have. In such systems, significant near-field effects were observed for Ag, while in the case of Au the effect is negligible (Cushing et al. J Phys Chem C 2015, 119, 16239). According to literature off-resonance excitation is able to generate significant local fields (Sun, et al. *Light* 2, e118 (2013); He et al. *Scie. Rep.* 6, 20777 (2016) & Hermann et al. *Opt. Exp.* 26, 27668 (2018)) but produce very low amount of useful hot carriers (Tagliabue, et al. *Nat. Commun.* 9, 3394 (2018)). This offered us the possibility to evaluate if the near-fields can affect directly the catalysis. Since this was found to not be the case, and the fact that one established that the ligands were reduced via unbiased ultrafast spectroscopies, it is possible to role near-fields as the corporate for the light-induced enhancement.

Action taken: we added to the revised text (including references):

It is clear from the data that the systems with plasmonic materials are responsive to the 532 nm illumination. Light-mediated plasmon-catalysis is a very complex process due to many potential reaction enhancers. One possibility is the near-field enhancements caused by the local electric fields formed upon Au LSPR excitation. At the most basic level near-fields can enhance charge separation and alignment of molecular dipoles. However, such localized electric fields impact the hot carriers but cannot catalyze the reaction autonomously. The second option employs the plasmon-induced resonance energy transfer (PIRET) process, connecting the plasmon evanescent field to a semiconductor absorber through dipole-dipole interaction. However, these systems necessitate core-

shell architectures (which are not applicable in this context), and the most substantial enhancements were observed with silver as the plasmonic material rather than gold. The final option explores strong-correlated plasmon-molecule systems, but in this scenario, there must be an optical overlap between the plasmon and molecule, which is once again not present in this case. To assess local field enhancement contribution, catalytic performance measurements were conducted using off-resonant excitation. Off-resonance excitation induces local effects such as elevated local near fields. However, in the context of hot carriers, off-resonance excitation generates low-energy carriers that are not conducive to driving photocatalytic processes. Excitation at 650 nm (off-resonant) caused no significant differences in the CV compared to experiments performed in the dark (Figure S17). This result was further supported by light switch chronoamperometry (Figure S18). The findings imply that local near-fields do not contribute significantly to enhancing catalysis. Consequently, the observed increase in catalytic output under resonant illumination is likely associated with hot electrons and heat rather than near fields. This, however, does not rule out the potential for near-fields to assist catalysis by enhancing charge separation; they would likely influence hot carriers indirectly engaged in the catalytic process.

2) The thermolysis of the bounded catalyst should be shown by external heating. What do you detect when the catalyst breaks down? This is the main support for discarding thermal effects and it should be better demonstrated.

We thank the Reviewer for the comment. Since the catalysis experiments were performed in an aqueous medium, it is impossible to perform thermolysis experiments at relevant temperatures (at least 500C). However, we can state that experiments performed at about 60C did not change significantly the catalytic performance.

The issue with plasmonic catalysis is that heat will always be present since we don't use all the charges. Additionally, surface temperature (much higher than the solution) is challenging since it also has ultrafast dynamics. Therefore, we performed light modulation experiments and analysed the shape catalytic response. This is presented in the new figure 3, and discussed in detail in the following comment. We think these additional experiments confirm the direct involvement of hot carriers as the prime culprit for the process while not discarding some positive contribution of heat.

Action taken: we added Figure 4 to the revised version of the manuscript and the text:

Heat is an inherent factor in plasmonic catalysis due to the occasional underutilization of charges, leading to their recombination and the generation of local heat. Although it is acknowledged that the surface temperature of excited plasmonic materials exceeds that of the solution, determining the precise value poses a challenge due to the ultrafast dynamics of thermalization. The molecular catalyst remains stable only up to 265 °C, a temperature considerably lower than what is required for uncatalyzed water thermolysis. Nevertheless, there exists a noteworthy temperature range that remains unexplored, primarily because experiments are conducted in an aqueous medium.

3) I disagree with the discussion regarding the wavelength-dependent experiments (S17 and S18) and the fact that they further support a hot-carriers pathway. Exciting the plasmon resonance is also the most efficient way to heat the system, so both processes, photothermal heating (that depends on the absorption cross-section) and hot-carriers generation are maximized at the same wavelength. For that reason, I think that point 2 is relevant (to show the thermolysis by external heating more than wavelength-dependence).

We thank the Reviewer for the comment. We agree that the off-resonant experiments cannot rule out the heat contribution since one expects low heat generation since we formed low energy carriers. However, the experiment provided evidence against the near-field enhancement.

To disentangle the heat contribution, we performed light modulation chronoamperometry. Maley et al. (J. Phys. Chem. C 2019, 123, 12390) showed that light absorption at electrode surfaces in nanoparticle arrays created significant local temperature increases and solution flows. These thermal effects were predicted to alter observed electrochemical currents through various mechanisms, including mass transfer enhancements, shifts in equilibrium redox potentials, or conventional temperature-dependent increases in kinetic rates for electrode processes. In particular, the presented analysis predicts that mass transfer enhancements alone would result in sizable current increases, and these enhancements would apply to any electrochemical reaction involving dissolved reactants and/or products. This was found valid for both outer-sphere and inner-sphere reactants. Consequently, heat-induced effects have a characteristic slow rise and decay of the current under light modulation since they operate on processes with time constants in the nanosecond time scale.

The light response of the complete system (NiO/Au-Co catalyst) versus the system without NiO (i.e. Au/Co-catalyst) provides clues for the contribution of heat to the process (figure 2C and D, compounded figure below)). By removing the NiO, the lifetime of the charge-separated state is expected to decrease and thus generate more heat. Consequently, if heat is the main contributor to the reactivity, one should expect a higher current when NiO is not present, which is not the case because we measured four times higher current induced by light when NiO was present. Furthermore, analysis of the response of the Au/Co-catalyst to light modulation shows a classic heat-mediated process with a relatively slow rise (see figure below) and decay, contrasting with the complete system which shows a faster rise and decay to the light modulation, indicating hot carrier involvement.

To decouple heat from hot carriers and further substantiate our mechanistic claim, we performed light modulation experiments with a higher repetition rate (new Figure 3). By increasing the repetition rate one expects lower heat accumulation at the electrode and thus clearer signal for hot carriers.

Action taken: we added Figure 4 to the revised version of the manuscript and the text:

Heat is an inherent factor in plasmonic catalysis due to the underutilization of hot charges, leading to their recombination and the generation of local heat. Although it is acknowledged that the surface temperature of excited plasmonic materials exceeds that of the solution, determining the precise value poses a challenge due to the ultrafast dynamics of thermalization. The molecular catalyst remains stable only up to 265 °C, a temperature considerably lower than what is required for uncatalyzed water thermolysis. Nevertheless, there exists a noteworthy temperature range that remains unexplored, primarily because experiments are conducted in an aqueous medium.

To disentangle the heat contribution, we performed light modulation chronoamperometry. In a study by Maley et al., it was demonstrated that light absorption at the electrode surfaces within nanoparticle arrays led to significant localized temperature increases and altered solution flows. These thermal effects were anticipated to influence electrochemical currents through diverse mechanisms, encompassing enhancements in mass transfer, shifts in equilibrium redox potentials, and conventional temperature-dependent accelerations in kinetic rates for electrode processes. Notably, their analysis suggests that mass transfer enhancements alone would result in substantial current increases applicable to electrochemical reactions involving dissolved reactants and products, both outer-sphere and inner-sphere reactants alike. Consequently, heat-induced effects exhibit a distinctive gradual rise and decay of the current during light modulation, as they operate on processes with time constants in the nanosecond range.

The light response of the entire system (NiO/Au-Co catalyst) in comparison to the system without NiO (i.e., Au/Co-catalyst) offers insights into the role of heat in the process (see figure 3C and D, compounded figure below). Removing NiO is expected to decrease the lifetime of the charge-separated state, generating more heat. However, contrary to the expectation that heat is the primary contributor to reactivity, we observed a fourfold increase in current induced by light when NiO was present. Additionally, examining the response of the Au/Co-catalyst to light modulation reveals a classic heat-mediated process with a relatively slow rise and decay, in contrast to the complete system. The complete system demonstrates a faster rise and decay to light modulation, indicating the involvement of hot carriers.

Light absorption by planar electrodes randomly decorated with plasmonic structures acts as a uniform heat source delocalized across the electrode-solution interface, resulting in heat dissipation in a linear geometry with significant temperature changes as a function of time. Consequently, the electrochemical response to light modulation provides a strategy to decouple heat from hot carriers' contributions to substantiate our mechanistic claim. Commonly, the experiments are performed by modulating the light intensity. However, the tested electrodes are quasi-transparent, making light-intensity modulation studies challenging. Thus, we opted to change the light ON/OFF cycle repetition rate to modulate electrode exposure to light.

Figure 4A shows the changes in measured photocurrent (Δi) as a function of light modulation repetition rate. Unsurprisingly, lower repetition rates (i.e. higher light exposure) resulted in larger Δi . In a heat-mediated process, the Δi is expected to scale with $\sim t^{1/2}$ (t = time), inconsistent with the observed current transients, providing the first substantiation for a hot carrier-mediated process. Additionally, Δi in a heat-mediated electrochemical process follows a linear dependence with increased light exposure, independent of the process occurring via inner or outer sphere reaction. Figure 4B shows that Δi does not show a linear behavior regarding light exposure, thus providing clearer evidence for hot carriers' involvement.

Figure 4. Light-modulated photo-electrocatalytic studies. The effect of light-modulation in the chronoamperometry was performed at -0.65 V vs Ag/AgCl with 3 mM acetic acid ($\text{pH} = 3.5$) and 532 nm CW laser with 43.8 mW/cm². The experiments were performed using squared function at different repetition rates. A) Changes in measured photocurrent (Δi) at different light modulation frequencies (8, 11, 16 and 33 mHz) over different light ON/OFF cycles; and B) Δi versus light modulation frequency.

As one of the key motivations of the paper is to disentangle the mechanisms behind the reported activity under plasmon excitation, I think it would be necessary to show in a more comprehensive way the role of field enhancement and heat.

We think the additional data and analysis substantiate our claim that hot electrons are involved in the process and are the main culprit for the light-induced response. The off-resonance measurements ruled out the near-field contribution as the direct cause of catalysis. However, they might be implicated in enhancing hot electrons' lifetime.

The new analysis of the response of the systems to light and the new light modulation experiments further substantiate the hot electron mechanism to the detriment of heat. Still, the heat was found to have some positive effect on the catalysis.

Action taken: we added Figure 4 to the revised version of the manuscript and the text:

In summary, a photosystem was proposed to confirm the direct involvement of hot electrons in a photocatalytic process, in this case H₂ evolution process. The photosystem effectively mitigates the heat contribution by designing a catalyst that decomposes well below water thermolysis conditions, positioning heat as a mere enhancement rather than the primary cause for the observed H₂ evolution. Off-resonance measurements conclusively eliminate near-field contributions as the direct catalyst of the reaction, although they may still play a role in extending the lifetime of hot electrons. The catalytic response to light modulation exhibits a shape consistent with the desired electron mechanism, contrasting with the detrimental impact of heat. Nevertheless, it was discovered that heat does have some positive influence on catalysis. Unbiased ultrafast spectroscopic measurements confirm charge transfer to respective acceptors. Additionally, in conjunction with NAP-XPS under variable potential, a postulated reaction mechanism highlights the crucial role of cobalt catalyst ligands. These ligands accept plasmon hot electrons and, through CPET steps, reduce and protonate the metal centre, ultimately leading to hydrogen evolution. This study conclusively resolves the longstanding debate within the research community regarding the direct involvement of hot carriers in the photocatalytic process.

REVIEWERS' COMMENTS

Reviewer #1 (Remarks to the Author):

I am now satisfied with the revision.

Reviewer #2 (Remarks to the Author):

I think the authors did a very good job in addressing the second round of questions. The new chronoamperometric results are highly appreciated to clarify the carriers versus heat issue. I think the manuscript is ready for acceptance.